# Neural Feature Geometry Evolves as Discrete Ricci Flow

**Moritz Hehl** [1 2]   **Max-K. von Renesse** [2]   **Melanie Weber** [3]

## Abstract

Deep neural networks learn feature representations via complex geometric transformations of the input data manifold. Despite the models' empirical success across domains, our understanding of neural feature representations is still incomplete. In this work we investigate neural feature geometry through the lens of discrete geometry. Since the input data manifold is typically unobserved, we approximate it using geometric graphs that encode local similarity structure. We provide theoretical results on the evolution of these graphs during training, showing that nonlinear activations play a crucial role in shaping feature geometry in feedforward neural networks. Moreover, we discover that the geometric transformations resemble a discrete Ricci flow on these graphs, suggesting that neural feature geometry evolves analogous to Ricci flow. This connection is supported by experiments on over 20,000 feedforward neural networks trained on binary classification tasks across both synthetic and real-world datasets. We observe that the emergence of class separability corresponds to the emergence of community structure in the associated graph representations, which is known to relate to discrete Ricci flow dynamics. Building on these insights, we introduce a novel framework for locally evaluating geometric transformations through comparison with discrete Ricci flow dynamics. Our experimental results further suggest potential connections between the evolution of feature geometry, and training time and network depth.[1]

## 1. Introduction

Deep neural networks have achieved remarkable success across diverse domains. Yet, a comprehensive theoretical understanding of why these models generalize and perform so well in practice remains elusive. To address this challenge, recent works have investigated how the geometry (Baptista et al., 2024; Ansuini et al., 2019; Cohen et al., 2020) and topology (Magai & Ayzenberg, 2022; Naitzat et al., 2020) of neural feature representations evolve as data propagates through network layers. Beyond advancing interpretability and explainability, such analyses also provide practical benefits, offering principled guidance for model and hyperparameter selection.

In this work we adopt a geometric perspective to analyze how deep neural networks evolve feature representations. Since the underlying manifold is not directly observable, we approximate its geometry by constructing geometric graphs from local similarity structure in the data. To the best of our knowledge, no prior work has provided theoretical results on how the geometry of such graphs evolves as data manifolds propagate through network layers. We provide initial theoretical insights by proving that, in the wide regime, deep linear networks preserve feature geometry, whereas nonlinear activations, such as ReLU, enable genuine geometric transformations.

Among the geometric concepts available for studying these transformations, Ricci curvature and its associated Ricci flow stand out as fundamental tools from Riemannian geometry. Originally introduced by Hamilton (1982), the Ricci flow intuitively describes the smoothing of a manifold's geometry through the evolution of its metric tensor. Famously, Perelman (2002; 2003b;a) employed it to prove the Poincaré conjecture and Thurston's geometrization conjecture. By carefully handling singularities, Perelman's work revealed topological insights through the progressive smoothing of the manifold's geometry. This mathematical framework bears a compelling analogy to deep neural networks, which progressively simplify and smooth the geometry of data manifolds, thereby uncovering richer information about the underlying classes in classification tasks.

Building on this intuition, we propose a novel framework for locally evaluating geometric transformations through comparison with discrete Ricci flow dynamics. We con-

---

[1]Max Planck Institute for Mathematics in the Sciences, Leipzig, Germany [2]Institute of Mathematics, Leipzig University, Leipzig, Germany [3]Department of Applied Mathematics, Harvard University, Cambridge, USA. Correspondence to: Melanie Weber <mweber@seas.harvard.edu>.

*Proceedings of the $43^{rd}$ International Conference on Machine Learning*, Seoul, South Korea. PMLR 306, 2026. Copyright 2026 by the author(s).

[1]Code available at https://github.com/Weber-GeoML/Ricci-Flow_Feature_Geometry

duct experiments on more than 20,000 feedforward neural networks trained on binary classification tasks across both synthetic and real-world datasets. We find that across datasets and architectures, neural networks consistently impose curvature-driven transformations closely aligned with the Ricci flow dynamics. Moreover, the emergence of class separability is reflected in the development of community structure in the associated graph representations, an evolution known to be closely tied to discrete Ricci flow dynamics (Tian et al., 2025; Ni et al., 2019; Lai et al., 2022).

Our experimental results suggest potential connections between the evolution of feature geometry, and training time and network depth. We find evidence that the emergence of geometrically informed feature transformations during training can inform early stopping. Additionally, by analyzing curvature-driven transformations layer-wise, we identify a critical point beyond which additional layers cease to yield meaningful curvature-driven changes. This suggests a relation between feature geometry and network depth selection.

Our proposed framework opens new avenues for understanding the geometric principles underlying deep learning that could inform practical tools for improving training efficiency and parameter selection across diverse applications.

**Summary of contributions** The main contributions of this work are as follows:

1. We prove that, in the wide regime, deep linear networks preserve feature geometry, whereas non-linear activations such as ReLU enable meaningful geometric transformations (Sec. 3.1). These results provide the mathematical foundation for our experimental analysis.

2. Our experiments indicate that the progressive emergence of class separability is reflected in the emergence of community structure within the corresponding graph representations (Sec. 4.2).

3. We provide experimental evidence that links the evolution of feature geometry to optimal training time selection (Sec. 4.3).

4. An analysis of layer-wise curvature-driven transformations suggests that the evolution of feature geometry relates to optimal network depth (Sec. 4.4).

**Related work** A variety of approaches have been proposed to better understand the feature transformations of deep neural networks. The connection between deep learning and Ricci flow was first explored by Baptista et al. (2024), who analyzed geometric transformations via Ricci flow at a global scale. Our approach differs by capturing the inherently local behavior of Hamilton's Ricci flow and by leveraging more refined discretizations of Ricci curvature.

Other efforts include topology-based analyses (Naitzat et al., 2020), and geometric measures of simplification (Brahma et al., 2015; Ansuini et al., 2019; Cohen et al., 2020). We defer a more detailed discussion of related literature to Appendix A.1.

## 2. Background and Notation

Following standard notation, we use $a$, $\boldsymbol{a}$, and $\boldsymbol{A}$ to denote scalars, vectors, and matrices. For $\mathbf{x} \in \mathbb{R}^n$, $\|\mathbf{x}\|$ denotes the $L^2$ norm. $\mathcal{N}(\mu, \sigma^2)$ represents a normal distribution with mean $\mu$ and variance $\sigma^2$. We denote a graph as $\mathcal{G} = (V, E)$, where $V$ is the vertex set and $E \subseteq V \times V$ the edge set. We write $u \sim v$ if $(u, v) \in E$ and $d(u, v)$ denotes the shortest path distance between $u$ and $v$. The 1-hop neighborhood of $v$ is denoted by $N(v) = \{u \in V : u \sim v\}$ and the degree by $\deg(v) = |N(v)|$. The maximum degree is given by $\deg_{\max} = \max_{v \in V} \deg(v)$.

### 2.1. Setting

To study the feature geometry of deep neural networks, we focus on binary classification, a fundamental task in supervised learning. Following the notation of Naitzat et al. (2020), we consider a compact manifold $M = M_a \cup M_b \subseteq \mathbb{R}^n$, given by the disjoint union of two submanifolds. The task is to determine, given a sample $\boldsymbol{x} \in M$, whether it belongs to $M_a$ or $M_b$.

To this end, we train a feed-forward neural network $\Phi : \mathbb{R}^n \rightarrow [0, 1]$ with $L$ hidden layers, given by

$$\Phi = \phi_{L+1} \circ \phi_L \circ \ldots \circ \phi_1.$$

Each layer of the network is defined as the composition of an affine transformation and a non-linear activation function $\sigma$, i.e., $\phi_\ell : \mathbb{R}^{n_{\ell-1}} \rightarrow \mathbb{R}^{n_\ell}$ is given by, $\phi_\ell(\boldsymbol{x}) = \sigma(\boldsymbol{W}_\ell \boldsymbol{x} + \boldsymbol{b}_\ell)$, where $\boldsymbol{W}_\ell \in \mathbb{R}^{n_\ell \times n_{\ell-1}}$ is the weight matrix and $\boldsymbol{b}_\ell \in \mathbb{R}^{n_\ell}$ is the bias vector. Here, $n_\ell$ denotes the width of layer $\ell$, with $n_0 = n$ corresponding to the input dimension. In this work, we use the ReLU activation function, defined as $\sigma(z) = \max(0, z)$, applied elementwise in all hidden layers. To produce probabilistic outputs, we apply a sigmoid activation in the final layer, i.e., $\phi_{L+1}(\boldsymbol{x}) = \rho(\boldsymbol{W}_{L+1} \boldsymbol{x} + \boldsymbol{b}_{L+1})$, where $\boldsymbol{W}_{L+1} \in \mathbb{R}^{1 \times n_\ell}$, $\boldsymbol{b}_{L+1} \in \mathbb{R}$ and $\rho(z) = \frac{1}{1+e^{-z}}$.

We study how the geometry of data evolves as it propagates through neural networks. Given an input manifold $M$, we denote by $\Phi_\ell = \phi_\ell \circ \ldots \circ \phi_1$ the composition of the first $\ell$ layers, and refer to $\Phi_\ell(M)$ as the feature manifold at layer $\ell$.

*Remark* 2.1. We follow standard terminology in referring to $M$ as a manifold and to $\Phi_\ell(M)$ as a feature manifold. Strictly speaking, neither assumption is essential for the following analysis. The manifold hypothesis serves as a heuristic motivation for using local geometric tools such as

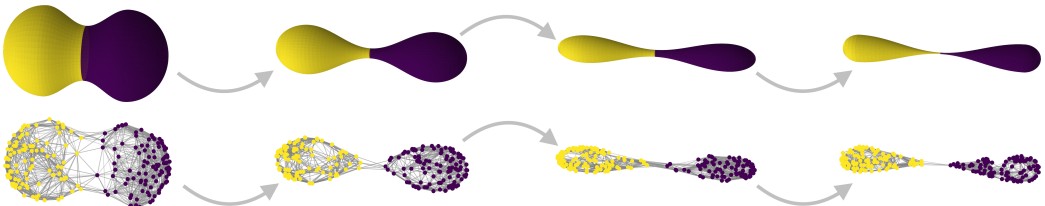

*Figure 1.* Schematic illustration of evolving feature manifolds (top row) along with the corresponding geometric graphs (bottom row) approximating their evolving geometry.

geometric graphs and discrete Ricci curvature. We note that natural data distributions are more accurately described as living near a low-dimensional tame geometric object, which is sometimes a manifold, and that the image $\Phi_\ell(M)$ under a non-invertible network with ReLU activations is in general not a manifold. Our theoretical and empirical results are properties of the graphs constructed from finite point clouds and do not require either set to admit a smooth manifold structure.

In practice, $M$ is unobserved, and we only have access to a finite set of samples $X = \{x^{(i)}\}_{i=1}^N \subset M$. To approximate the geometry of the feature manifolds, we construct geometric graphs on the transformed samples $\{\Phi_\ell(x^{(i)})\}_{i=1}^N$, as schematically illustrated in Figure 1. Graphs based on local connectivity patterns, such as $k$-nearest neighbor graphs or $r$-neighborhood graphs, are known to preserve geometric and topological properties of the manifold when samples are sufficiently dense, including Ricci curvature (Van Der Hoorn et al., 2021; García Trillos & Weber, 2026). This approach is well-established in manifold learning and geometric data analysis, where such graph-based representations are commonly used to study the geometry of data.

Specifically, we consider the $k$-nearest neighbor graph, denoted by $\mathcal{G}_k(X)$, where the vertices of $\mathcal{G}_k(X)$ correspond exactly to the samples in $X$, and two vertices are connected if either is among the $k$-nearest neighbors of the other, i.e., a symmetric $k$-NN graph. Additionally, we construct $r$-neighborhood graphs $G_r(X)$, where an edge is drawn between two vertices if their distance is less than a fixed radius $r > 0$. These graphs provide discrete approximations of the evolving feature manifolds.

### 2.2. Ricci Curvature of Graphs

Ricci curvature plays a fundamental role in Riemannian geometry and provides the foundation for our analysis of feature geometry. To extend curvature concepts to graphs, we adopt two of the most widely used discretizations, proposed by Ollivier (2009) and Forman (2003). We briefly introduce them below.

Intuitively, Ricci curvature measures how the local geometry of a manifold deviates from being flat. This can be captured

by comparing the distance between two nearby points with the distance between small geodesic balls centered at them: in regions of positive (negative) curvature, the geodesic balls are closer together (farther apart) than the points themselves.

Building on this intuition, Ollivier (2009) extends the classical notion of Ricci curvature to graphs by replacing geodesic balls with the transition probability of a random walk. For a vertex $u$, let $\mu_u$ denote the uniform distribution over its neighbors, i.e., $\mu_u(v) = \frac{1}{\deg(u)}$ if $u \sim v$ and $\mu_u(v) = 0$ otherwise. Ollivier-Ricci curvature then compares the distance between these distributions to the distance between their centers, mirroring the comparison between geodesic balls and their centers in the Riemannian case:

$$\mathcal{O}(u,v) = 1 - \frac{W_1(\mu_u, \mu_v)}{d(u,v)},$$

where $W_1(\mu_u, \mu_v)$ is the 1-Wasserstein distance, defined by

$$W_1(\mu_u, \mu_v) = \inf_{\pi \in \Pi(\mu_u, \mu_v)} \sum_{a \in V} \sum_{b \in V} d(a,b)\pi(a,b),$$

and $\Pi(\mu_u, \mu_v)$ denotes the set of all couplings of $\mu_u$ and $\mu_v$.

Computing Ollivier-Ricci curvature is computationally demanding, as it requires solving an optimal transport problem for each edge with complexity $O(\deg_{\max}^3)$ via the Hungarian algorithm. This can be mitigated by approximating the Wasserstein distance using Sinkhorn distances (Cuturi, 2013) or through direct combinatorial approximations of the Ollivier-Ricci curvature (Tian et al., 2025). We adopt the latter, detailed in Appendix A.2.1, in our experiments.

On the other hand, Forman (2003) introduced a discretization of Ricci curvature on CW complexes via a discrete analogue of the Bochner–Weitzenböck formula. For a simple, unweighted graph, the Forman-Ricci curvature of an edge $u \sim v$ is defined as

$$\mathcal{F}(u,v) = 4 - \deg(u) - \deg(v).$$

While this definition is well-founded in Forman's framework and computationally efficient, it is often too simplistic to capture the geometric complexity required in many applications. To address this limitation, augmented versions

of Forman's curvature have been considered (Bloch, 2014; Samal et al., 2018; Weber et al., 2018). A widely used refinement incorporates contributions from three-cycles, yielding the following combinatorial expression:

$$\mathcal{AF}(u,v) = 4 - \deg(u) - \deg(v) + 3|N(u) \cap N(v)|.$$

This augmentation can be computed in $O(E \deg_{\max})$ time, providing a scalable alternative to the computationally demanding Ollivier–Ricci curvature. A more detailed introduction to the Forman-Ricci curvature is provided in Appendix A.2.2.

### 2.2.1. CURVATURE GAP

When two adjacent vertices belong to the same community, their neighborhoods tend to be more tightly connected. This lowers the transport cost between neighborhood distributions, yielding higher Ollivier-Ricci curvature, and likewise increases augmented Forman-Ricci curvature due to a higher incidence of triangles. Both measures are therefore effective for community detection (Sia et al., 2019; Gosztolai & Arnaudon, 2021; Fesser et al., 2024). By contrast, the original Forman–Ricci curvature depends only on endpoint degrees and cannot reliably distinguish intra- from inter-community edges. As a result, Ollivier- and augmented Forman–Ricci curvature show a bimodal distribution in graphs with strong community structure. To quantify this bimodality, we use the curvature gap (Gosztolai & Arnaudon, 2021):

$$\Delta\mathcal{O} = \frac{1}{\sigma}\left(\mathcal{O}_{\text{intra}} - \mathcal{O}_{\text{inter}}\right)$$

where $\mathcal{O}_{\text{intra}}$ and $\mathcal{O}_{\text{inter}}$ denote the mean curvature of intra- and inter-community edges, and $\sigma$ is the pooled standard deviation. This measure captures how strongly the local graph geometry, as encoded by Ricci curvature, reflects community structure. The curvature gap can be analogously defined for augmented Forman–Ricci curvature. Visualizations and further community structure metrics (modularity, normalized cut, spectral gap) are presented in Appendix A.2.3.

### 2.3. Ricci Flow

To analyze the evolving geometry of the feature manifolds, it is natural to draw inspiration from the Ricci flow, a central concept in Riemannian geometry introduced by Hamilton (1982). The Ricci flow evolves a Riemannian metric $g$ according to $\frac{\partial}{\partial t}g(t) = -2\text{Ric}(g(t))$ with initial condition $g(0) = g$, where $\text{Ric}(g(t))$ denotes the Ricci curvature tensor. Further details on Ricci flow and its discretizations are provided in Appendix A.2.4. This evolution is often compared to heat diffusion, as the underlying equation shares a similar averaging effect, smoothing out curvature irregularities by shrinking positively curved regions and expanding negatively curved ones. While there is no unique notion

of discrete Ricci flow on graphs, the existing formulations, with the first being proposed by Ollivier (2010), all build on this fundamental geometric mechanism: positive curvature drives contraction and negative curvature drives expansion. Our analysis below targets this shared qualitative behavior rather than tracking any specific discretization, making our framework agnostic to the choice of discrete formulation.

## 3. Approximating Feature Geometry

This section establishes theoretical results on feature manifold evolution in wide neural networks, emphasizing the key role of non-linear activations in geometric transformations. We then introduce a novel measure that compares local network-induced geometric changes with those predicted by discrete Ricci flow.

### 3.1. Theoretical Results

As a first result, we show that for randomly initialized, sufficiently wide neural networks without nonlinearity, the graph structures encoding the feature geometry are preserved with high probability. Two graphs $G$ and $H$ are said to be *isomorphic*, denoted by $G \cong H$, if there exists a bijection between their vertex sets that preserves adjacency relations, i.e., the graphs are identical up to vertex relabeling. The following theorem establishes explicit lower bounds on the network width that guarantee the existence of an isomorphism between the $k$-nearest neighbor graphs.

**Theorem 3.1.** *Let $X \subset \mathbb{R}^n$ be a finite set, and assume there exists $0 < \epsilon < 1$ such that*

$$\min_{\substack{Y \subset X\backslash\{x\} \\ |Y|=k}} \max_{y \in Y}\|\boldsymbol{x}-\boldsymbol{y}\|^2 \leq \frac{1-\epsilon}{1+\epsilon} \min_{\substack{Y \subset X\backslash\{x\} \\ |Y|=k+1}} \max_{y \in Y}\|\boldsymbol{x}-\boldsymbol{y}\|^2$$

*for all $\boldsymbol{x} \in X$. Furthermore, let $\boldsymbol{A} \in \mathbb{R}^{m \times n}$ be a random matrix with i.i.d. entries $A_{ij} \sim \mathcal{N}(0, 1/m)$. Then, the map $\psi : X \mapsto \boldsymbol{A}X := \{\boldsymbol{A}\boldsymbol{x} : \boldsymbol{x} \in X\}$, defined by $\psi(\boldsymbol{x}) = \boldsymbol{A}\boldsymbol{x}$, is a graph isomorphism between $\mathcal{G}_k(X)$ and $\mathcal{G}_k(\boldsymbol{A}X)$ with probability bounded from below*

$$\mathbb{P}\left(\mathcal{G}_k(X) \cong \mathcal{G}_k(\boldsymbol{A}X) \text{ under } \psi\right)$$
$$\geq 1 - |X|(|X|-1)e^{\frac{m}{4}(\epsilon^3-\epsilon^2)}.$$

The condition on $\epsilon$ ensures that, for each point $x$, there is a strict gap between the distance to its $k$-th and $(k+1)$-th nearest neighbors, so that neighborhood membership is unambiguous. The ratio $(1-\epsilon)/(1+\epsilon)$ measures the size of this gap, with larger $\epsilon$ corresponding to a more robust separation between neighbors and non-neighbors. This makes the $k$-NN graph well-defined and stable under small perturbations. The condition is generically satisfied: for points drawn from a continuous distribution, distance ties occur with probability zero, so some valid $\epsilon > 0$ exists almost surely.

*Remark* 3.2. Since the addition of a bias term does not affect pairwise distances, the same result holds for one-layer linear networks with bias.

The proof builds on the Johnson–Lindenstrauss Lemma, which implies that randomly initialized weight matrices act as approximate isometries with high probability. The complete proof of Theorem 3.1 is deferred to Appendix A.3.1. Analogous results for $r$-neighborhood graphs (Theorem A.7), generalizations to deep networks (Theorem A.8), and empirical validation (Appendix A.4.1) are also provided.

Random initialization combined with over-parameterization keeps network weights near their initial values during gradient descent. We show that, without nonlinearities, network dynamics cannot alter the feature geometry encoded by graph structures, regardless of the number of gradient descent steps. Consider a two-layer network $\Phi = \phi_2 \circ \phi_1$ with $\phi_1(\boldsymbol{x}) = \sigma\left(\frac{1}{\sqrt{m}}\boldsymbol{W}\boldsymbol{x}\right)$, where $\sigma$ denotes the ReLU activation and $m$ the width of the hidden layer. We minimize the empirical loss by keeping the second-layer weights fixed, while gradient descent updates the first-layer weight matrix $\boldsymbol{W}$, denoted by $\boldsymbol{W}(l)$ after $l$ gradient descent steps. Then, the $k$-nearest neighbor graphs remain invariant prior to the nonlinearity, as stated in the following theorem.

**Theorem 3.3** (Informal). *Let $X \subset \mathbb{R}^n$ be a finite set. Under suitable technical assumptions, for networks of sufficient width $m$ and any number of gradient descent steps $l \geq 0$, the map*

$$\psi : X \to X(l) := \left\{ \frac{1}{\sqrt{m}}\boldsymbol{W}(l)\boldsymbol{x} : \boldsymbol{x} \in X \right\};$$

$$\psi(\boldsymbol{x}) = \frac{1}{\sqrt{m}}\boldsymbol{W}(l)\boldsymbol{x}$$

*is a graph isomorphism between $\mathcal{G}_k(X)$ and $\mathcal{G}_k(X(l))$ with high probability.*

A formal version of this result, including exact lower bounds on the required network width and the full proof, is provided in Appendix A.3.2. There, we also present an analogous theorem for $r$-neighborhood graphs.

The results above establish that wide linear neural networks cannot alter the underlying feature geometry, as their weight matrices act as approximate isometries. In contrast, once a nonlinearity is introduced, our experiments show clear changes in the geometry, as captured by the graph structures (see Section 4). This highlights the essential role of the ReLU activation in enabling such transformations. Building on this observation, we further demonstrate that even when the weight matrices are exact isometries, adding the ReLU nonlinearity is sufficient to change the geometry of the feature manifolds.

**Theorem 3.4** (Informal). *For any three vertices, there exists a linear isometry such that composing it with a ReLU activation changes the ordering of their pairwise distances. In particular, this operation can rewire the $k$-nearest neighbor graph.*

This provides not only empirical but also theoretical evidence for the fundamental role of the activation function in changing the feature geometry. A formal treatment of this result is provided in Appendix A.3.3.

### 3.2. Local Ricci evolution coefficients

In this section, we introduce a novel framework to evaluate the geometric changes induced by deep neural networks by drawing an analogy with the Ricci flow. Recall that the Ricci flow regularizes the geometry of a manifold by shrinking regions of positive curvature and expanding regions of negative curvature. We aim to assess whether neural networks induce feature transformations that exhibit a similar curvature-driven regularization. As there exists no canonical discretization of Ricci flow on graphs, we do not commit to any specific flow equation. Instead, we design a measure that captures the core qualitative signature of Ricci flow — the negative correlation between curvature and local expansion — making our framework agnostic to the choice of discrete formulation.

Since the feature manifolds cannot be directly observed, we instead approximate their geometry using the $k$-nearest neighbor graph $\mathcal{G}_k(\Phi_\ell(X))$, constructed from the transformed samples $\Phi_\ell(X) = \{\Phi_\ell(\boldsymbol{x}^{(i)})\}_{i=1}^N$ after layer $\ell$. A discussion on the choice of the parameter $k$ is provided in Appendix A.5.1.

To reflect the local nature of the Ricci flow in our graph-based framework, we focus on the smallest neighborhoods, i.e., the one-hop neighborhoods. The curvature of a one-hop neighborhood centered at a vertex $\boldsymbol{x}$ at layer $\ell$ is approximated by the discrete scalar curvature of Ollivier (2010),

$$\mathcal{O}_\ell(\boldsymbol{x}) = \frac{1}{\deg_\ell(\boldsymbol{x})} \sum_{\boldsymbol{y} \in N_\ell(\boldsymbol{x})} \mathcal{O}(\boldsymbol{x}, \boldsymbol{y}),$$

where $\deg_\ell(\boldsymbol{x})$ and $N_\ell(\boldsymbol{x})$ denote the degree and one-hop neighborhood of $\boldsymbol{x}$ in $\mathcal{G}_k(\Phi_\ell(X))$. To capture how a local region evolves across layers, we define the average change in distances

$$\eta_\ell(\boldsymbol{x}) = \frac{1}{\deg_\ell(\boldsymbol{x})} \sum_{\boldsymbol{y} \in N_\ell(\boldsymbol{x})} (d_{\ell+1}(\boldsymbol{x}, \boldsymbol{y}) - d_\ell(\boldsymbol{x}, \boldsymbol{y})),$$

where $d_\ell(\boldsymbol{x}, \boldsymbol{y})$ is the distance between $\boldsymbol{x}$ and $\boldsymbol{y}$ at layer $\ell$. Ideally, we would use geodesic distances on the underlying manifold; since the manifold is unobservable, we instead use the shortest path distances in the $k$-NN graph as

a discrete analog. Intuitively, $\eta_\ell(\boldsymbol{x})$ measures whether the neighborhood of $\boldsymbol{x}$ expands during the transition from layer $\ell$ to $\ell + 1$. Under the Ricci flow, positively curved regions contract while negatively curved regions expand, implying a negative correlation between $\mathcal{O}_\ell(\boldsymbol{x})$ and $\eta_\ell(\boldsymbol{x})$. To quantify this, we compute the Pearson correlation coefficient across layers,

$$\rho(\boldsymbol{x}) = \frac{\sum_{\ell=1}^{L-1}(\eta_\ell(\boldsymbol{x}) - \bar{\eta}(\boldsymbol{x}))(\mathcal{O}_\ell(\boldsymbol{x}) - \bar{\mathcal{O}}(\boldsymbol{x}))}{\sqrt{\sum_{\ell=1}^{L-1}(\eta_\ell(\boldsymbol{x}) - \bar{\eta}(\boldsymbol{x}))^2}\sqrt{\sum_{\ell=1}^{L-1}(\mathcal{O}_\ell(\boldsymbol{x}) - \bar{\mathcal{O}}(\boldsymbol{x}))^2}},$$

where $\bar{\eta}(\boldsymbol{x}) = \frac{1}{L-1}\sum_{\ell=1}^{L-1}\eta_\ell(\boldsymbol{x})$ and $\bar{\mathcal{O}}(\boldsymbol{x}) = \frac{1}{L-1}\sum_{\ell=1}^{L-1}\mathcal{O}_\ell(\boldsymbol{x})$ denote the averages across layers. We refer to $\rho(\boldsymbol{x})$ as the *local Ricci evolution coefficient* of the network at point $\boldsymbol{x}$. Although introduced here in the context of Ollivier curvature, the framework is general and can likewise be instantiated with alternative notions of discrete curvature, such as the augmented Forman curvature or efficient approximations of Ollivier curvature.

*Remark* 3.5. Appendix A.1.1 provides a detailed comparison between our local framework and the global approach of Baptista et al. (2024).

In addition to evaluating Ricci flow–like behavior at the level of individual neighborhoods, we can also assess it layer by layer. Specifically, we ask whether the geometric transformations induced by a given layer $\ell$ align with those expected under the Ricci flow. To this end, we define the *layer Ricci coefficient*

$$\rho(\ell) = \frac{\sum_{\boldsymbol{x}\in\Phi_\ell(X)}(\eta_\ell(\boldsymbol{x}) - \bar{\eta}_\ell)(\mathcal{O}_\ell(\boldsymbol{x}) - \bar{\mathcal{O}}_\ell)}{\sqrt{\sum_{\boldsymbol{x}\in\Phi_\ell(X)}(\eta_\ell(\boldsymbol{x}) - \bar{\eta}_\ell)^2}\sqrt{\sum_{\boldsymbol{x}\in\Phi_\ell(X)}(\mathcal{O}_\ell(\boldsymbol{x}) - \bar{\mathcal{O}}_\ell)^2}},$$

where $\bar{\eta}_\ell = \frac{1}{|X|}\sum_{\boldsymbol{x}\in\Phi_\ell(X)}\eta_l(\boldsymbol{x})$ and $\bar{\mathcal{O}}_\ell = \frac{1}{|X|}\sum_{\boldsymbol{x}\in\Phi_\ell(X)}\mathcal{O}_\ell(\boldsymbol{x})$. This coefficient provides a global summary of whether the geometric transformation induced by layer $\ell$ follows the curvature-driven dynamics of the Ricci flow.

A natural concern is how this framework scales to higher-dimensional data. The computational cost of discrete Ollivier-Ricci curvature depends on the structure of the $k$-NN graph, in particular the maximum degree, rather than on the ambient dimension. However, constructing a representative $k$-NN graph requires sufficient sampling density relative to the intrinsic dimension of the data manifold. The key theoretical requirement underlying our framework is the consistency between continuous and discrete Ricci curvature, for which García Trillos & Weber (2026) provide non-asymptotic guarantees whose sample complexity scales with the intrinsic rather than the ambient dimension. Since the intrinsic dimensionality of intermediate feature representations is typically much lower than the ambient dimension, and has been observed to decrease in deeper layers (Ansuini et al., 2019), we expect these scaling concerns to be mitigated in practice.

# 4. Experimental Analysis

## 4.1. Local Ricci evolution coefficients

Using our framework of local Ricci evolution coefficients, we empirically examine whether deep neural networks exhibit curvature-driven dynamics in the evolution of their feature geometry. To this end, we study both synthetic and real-world datasets. The synthetic datasets are constructed to span varying degrees of geometric and topological entanglement. For real-world benchmarks, we consider visually similar digit pairs from MNIST (1 vs. 7, 6 vs. 9), fine-grained visual distinctions from Fashion-MNIST—sneakers vs. sandals (FMNIST-SvS) and shirts vs. dresses (FMNIST-SvD)—and from CIFAR-10 (cars vs. planes). Further details on datasets and task setup are provided in Appendix A.5. We train feed-forward networks with varying widths and depths, all of which achieve over 99% training accuracy, ensuring that our analysis reflects meaningful learned feature representations. To account for randomness in training, results are averaged over 50 independently initialized and trained networks per dataset–architecture pair. In total, we analyze the feature geometry of more than 20,000 networks.

Table 1 reports results on real-world datasets, consistently showing negative local Ricci evolution coefficients, providing strong evidence of Ricci flow–like dynamics in feature geometry. The large majority of vertices exhibit negative coefficients, indicating that curvature-driven dynamics are a global phenomenon on the data manifold. To reduce computational overhead, we further compute local Ricci evolution coefficients using augmented Forman curvature and the approximate Ollivier curvature of Tian et al. (2025). Both yield results consistent with the exact Ollivier curvature while being substantially more efficient (see Tables 4 and 5). For completeness, we present the entire distribution of local Ricci evolution coefficients in Appendix A.4.2, along with results on synthetic datasets. Strikingly, we observe qualitatively identical behavior across all architectures and datasets, both synthetic and real, underscoring the robustness and universality of this phenomenon. Additionally, we calculated the local Ricci evolution coefficients using the Spearman correlation instead of the Pearson correlation. Since the Spearman correlation captures monotonic relationships, it is less sensitive to outliers or non-normal distributions. The results are presented in Table 10, and are closely aligned with the results using the Pearson correlation. Together, these findings provide compelling evidence that the evolution of feature geometry in deep neural networks is fundamentally curvature-driven, closely aligned with Ricci flow.

## 4.2. Community structure

We study graphs whose nodes can be naturally partitioned into two communities according to the true labels of the underlying binary classification task. This setup is well

*Table 1.* Average local Ricci evolution coefficients on real-world data. Values are means $\pm$ standard deviations over 50 independently trained networks per architecture; proportion of vertices with negative coefficients is reported in parentheses. Networks were randomly initialized.

| (Width,Depth) | MNIST-1v7 | MNIST-6v9 | FMNIST-SvS | FMNIST-SvD | CIFAR |
|---|---|---|---|---|---|
| (15, 7) | $-0.58 \pm 0.08$ (88.7%) | $-0.51 \pm 0.09$ (85.3%) | $-0.43 \pm 0.05$ (84.0%) | $-0.27 \pm 0.08$ (73.4%) | $-0.44 \pm 0.12$ (87.8%) |
| (15, 10) | $-0.60 \pm 0.06$ (91.8%) | $-0.59 \pm 0.06$ (92.6%) | $-0.40 \pm 0.05$ (84.4%) | $-0.29 \pm 0.12$ (77.6%) | $-0.43 \pm 0.15$ (87.8%) |
| (15, 15) | $-0.61 \pm 0.07$ (93.3%) | $-0.58 \pm 0.11$ (92.9%) | $-0.52 \pm 0.11$ (93.8%) | $-0.40 \pm 0.12$ (88.2%) | $-0.55 \pm 0.18$ (93.3%) |
| (25, 7) | $-0.58 \pm 0.05$ (89.3%) | $-0.48 \pm 0.10$ (83.3%) | $-0.41 \pm 0.03$ (81.9%) | $-0.28 \pm 0.08$ (74.3%) | $-0.48 \pm 0.13$ (89.9%) |
| (25, 10) | $-0.62 \pm 0.05$ (92.8%) | $-0.59 \pm 0.05$ (92.8%) | $-0.40 \pm 0.05$ (84.8%) | $-0.32 \pm 0.09$ (80.4%) | $-0.54 \pm 0.13$ (94.8%) |
| (25, 15) | $-0.60 \pm 0.06$ (94.2%) | $-0.61 \pm 0.07$ (94.9%) | $-0.47 \pm 0.08$ (93.5%) | $-0.46 \pm 0.08$ (92.2%) | $-0.71 \pm 0.06$ (98.1%) |
| (50, 7) | $-0.59 \pm 0.05$ (90.6%) | $-0.46 \pm 0.14$ (82.0%) | $-0.42 \pm 0.03$ (83.0%) | $-0.35 \pm 0.09$ (80.8%) | $-0.57 \pm 0.12$ (95.4%) |
| (50, 10) | $-0.65 \pm 0.04$ (94.6%) | $-0.61 \pm 0.07$ (93.3%) | $-0.43 \pm 0.07$ (86.5%) | $-0.44 \pm 0.10$ (88.8%) | $-0.70 \pm 0.05$ (98.5%) |
| (50, 15) | $-0.63 \pm 0.06$ (95.2%) | $-0.61 \pm 0.08$ (95.0%) | $-0.54 \pm 0.05$ (96.0%) | $-0.53 \pm 0.07$ (95.0%) | $-0.76 \pm 0.04$ (98.3%) |

*Figure 2.* Modularity and normalized cut across network layers on synthetic datasets. Reported values are averaged over 50 independently trained networks with random initialization and one standard deviation is shown as envelopes around the mean.

suited for a community-detection perspective. In this section, we examine whether the class separability learned by deep neural networks induces a rewiring that strengthens the community structure of the $k$-nearest neighbor graphs.

To this end, we evaluate how well the geometry of the graphs aligns with the prescribed community structure by measuring the curvature gap, modularity, and normalized cut. Our experiments on both synthetic and real-world datasets show that the community structure becomes increasingly pronounced as the networks evolve the feature geometry. Figure 2 reports the evolution of modularity and normalized cut across network layers, averaged over 50 independently trained models to mitigate stochastic variability. In all datasets, we observe a consistent increase in modularity and a corresponding decrease in normalized cut, indicating that the learned feature geometry progressively aligns with the prescribed community structure. For real-world datasets, this effect is still present but less pronounced, as the $k$-nearest neighbor graphs constructed from raw inputs already exhibit relatively high modularity, particularly in the case of MNIST (see Figure 10).

In our setting, the curvature gap does not reliably capture how well the graph geometry aligns with the prescribed community structure. Most inter-community edges arise from misclassified nodes connected to correctly classified ones with the same label, which the network effectively treats as intra-community edges, making them indistinguishable through the curvature lens. To clarify this effect, Figure 3 compares the curvature gaps on the MNIST 1-vs-7 dataset computed on the full test set with those computed after re-

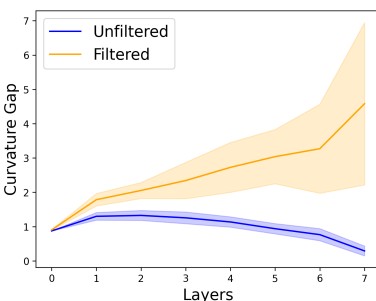

*Figure 3.* Curvature gaps before and after removing misclassified samples.

moving the five misclassified points (out of 1000). While removing such a small fraction of samples should not noticeably alter the graph geometry, it leads to a qualitatively different behavior: the curvature gap increases consistently across layers instead of collapsing. This is expected, as inter-community edges now differ structurally from intra-community ones. We discuss this phenomenon in more detail in Appendix A.4.4.

Overall, these results demonstrate that deep neural networks progressively evolve the geometry of feature manifolds in a manner that amplifies the underlying community structure.

### 4.3. Overfitting and local Ricci evolution coefficients

To better understand how neural networks learn the geometry of the data manifold, we track the local Ricci evolution coefficients during training. Across all datasets, we observe a strikingly consistent pattern: at the beginning of training,

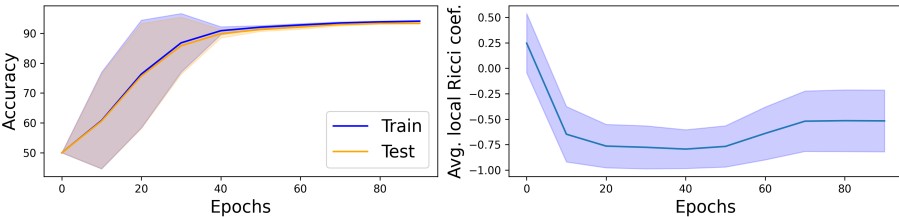

*Figure 4.* Average local Ricci evolution coefficients, computed from the approximated Ollivier–Ricci curvature, shown with the corresponding accuracies throughout training on the Fashion-MNIST dataset. Reported values are averaged over 50 independently trained networks with random initialization.

the mean coefficients exhibit a sharp decline, suggesting that the network is effectively learning the underlying geometric structure. Once test accuracy stabilizes, however, this trend reverses: the mean coefficients plateau or rise again. We hypothesize that this marks a shift in training dynamics, where the network ceases to capture new geometric structure and instead begins to overfit individual samples. This pattern suggests that monitoring local Ricci evolution coefficients during training could serve as a principled stopping heuristic. In practice, this can be made more efficient by approximating Ollivier–Ricci curvature or by using augmented Forman curvature, both of which lower computational cost while retaining the essential geometric signal. Figure 4 illustrates this phenomenon on the Fashion-MNIST dataset, showing the local Ricci evolution coefficients alongside train and test accuracy throughout training.

To further test this interpretation in a setting where overfitting is induced by design, we conducted additional experiments on a synthetic dataset with 25% of the training labels flipped. The resulting dynamics are again consistent with our hypothesis. Details are provided in Appendix A.4.3.

### 4.4. Analysis across layers

We now turn to the evaluation of the layer-Ricci coefficients, introduced in Section 3.2. We compute these coefficients across both synthetic and real-world datasets, considering networks of varying depth, while keeping the width fixed. As before, all models are trained to exceed $99\%$ training accuracy to ensure that we analyze meaningful learned representations. For each dataset-architecture pair, results are averaged over 50 independently trained networks to account for stochasticity in initialization and optimization.

Across all experiments, we observe a strikingly consistent behavior: the curves of the layer-Ricci coefficients follow the same trend across network depths. Specifically, there appears to be a critical depth up to which the coefficients decrease, and after which they begin to increase again. This turning point suggests a balance between the network's ability to capture geometric structure and its tendency to overfit. Up to the critical depth, additional layers appear to enrich

the evolution of the feature geometry, as reflected by decreasing Ricci coefficients. Beyond this point, however, further depth no longer contributes meaningful geometric transformations, which manifests as increasing Ricci coefficients. This phenomenon highlights the critical depth as a potential heuristic for selecting network architectures: it indicates the point at which adding more layers ceases to provide geometric benefits. An example of this behavior on the MNIST dataset is shown in Figure 5. Notably, the depth identified by this procedure coincides with the depth that maximizes test accuracy when averaged over 50 independently trained networks.

We additionally explored whether the same depth-dependent behavior occurs in ResNet architectures, which are known to benefit from greater depth. Repeating the analysis on ResNets under the same experimental setup, we find that layer-Ricci coefficients remain consistently negative and follow the same qualitative trend across depths. In contrast to standard feed-forward networks, where the coefficients rise beyond a critical depth, those of ResNets continue to decrease with additional layers, in line with the well-known capacity of ResNets to leverage depth more effectively. Figure 6 illustrates this on the MNIST 1-vs-7 dataset. These results suggests that the layer-Ricci framework captures architecture-dependent differences in how depth contributes to the geometric evolution of features.

## 5. Discussion

**Summary**  In this paper we have introduced the *local Ricci evolution coefficients*, a tool to evaluate locally the geometric transformations of feature manifolds by comparing them to Ricci flow dynamics. We theoretically show that nonlinear activations are essential for reshaping feature geometry. Empirically, we demonstrate that the progressive emergence of class separability is mirrored in the development of community structure within the corresponding graph representations. Moreover, our experiments indicate that well-trained networks exhibit curvature-driven transformations closely aligned with Ricci flow, and that this behavior emerges during training. We further provide experimental evidence for connections to optimal training time and depth selection.

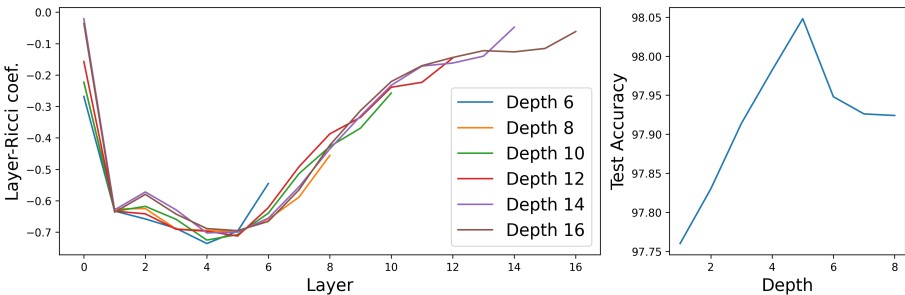

*Figure 5.* Layer-Ricci coefficients, computed from the augmented Forman-Ricci curvature, on the MNIST 1-vs-7 dataset for networks of varying depth (width fixed to 25). Reported values are averaged over 50 independently trained networks with random initialization.

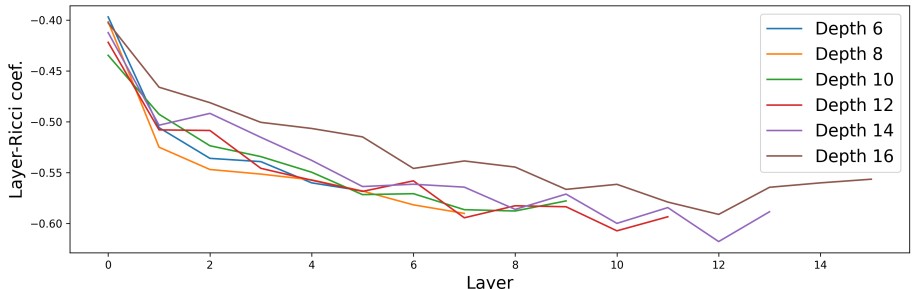

*Figure 6.* Layer-Ricci coefficients, computed from the augmented Forman-Ricci curvature, on the MNIST 1-vs-7 dataset for ResNets of varying depth (width fixed to 25). Reported values are averaged over 50 independently trained networks with random initialization.

**Limitations and future work**  Our use of the term "manifold" is heuristic. The manifold hypothesis motivates our use of local geometric tools, but our results concern graphs built from finite point clouds and do not rely on the data or its feature representations being supported on a smooth manifold.

The theoretical results in Section 3.1 are not intended to explain why neural networks exhibit Ricci flow-like dynamics, a challenging open problem, but rather to establish that such dynamics are only possible in the presence of non-linearities, thereby providing the mathematical foundation for the empirical analysis in Section 4.

While we have established the importance of non-linear activations in reshaping feature geometry, deriving exact evolution equations for graphs constructed from local connectivity patterns in non-linear networks remains an open problem. Moreover, our study was conducted on relatively small datasets and focused on feed-forward architectures; extending the analysis to larger-scale datasets and more diverse architectures (e.g., convolutional neural networks) as well as kernel-based methods represents a valuable direction for future work.

Another interesting avenue for future study is to analyze the double descent phenomenon (Belkin et al., 2019) through the framework of local Ricci evolution coefficients. In the overparameterized regime, our results show that increasing network size—either by expanding depth at fixed width or width at fixed depth—systematically raises the proportion of vertices with negative Ricci coefficients. This suggests that larger networks operate in a more geometry-aware manner, providing a novel geometric perspective on the mechanisms underlying double descent. Further discussion and initial experimental results can be found in Appendix A.4.5. The connections between the evolution of feature geometry, and training time and network depth suggest heuristics for optimal stopping and optimal choice of the number of layers. A systematic investigation of these heuristics is an important direction for future work. Furthermore, local Ricci evolution coefficients could serve as a novel tool to detect geometric anomalies and support uncertainty quantification in deep neural networks, since regions of the data manifold with non-negative coefficients may signal unexpected geometric behavior by the network. Additionally, investigating our curvature-driven geometric measures in conjunction with node-level graph curvatures, such as Bakry-Émery or resistance curvature, offers a particularly interesting avenue for future study. Finally, while the relationship between intrinsic dimensionality and the geometric measures introduced in this work remains unclear to us, we consider this an especially intriguing question to investigate in future research.

## Impact Statement

This paper presents work whose goal is to advance the field of Machine Learning. There are many potential societal consequences of our work, none which we feel must be specifically highlighted here.

## Acknowledgements

MR thanks Karl Schlagenhauf for a very inspiring breakfast conversation triggering his interest in the question. MH wants to thank Willem Diepeveen for fruitful discussions and Guido Montúfar for his support. MH was partly supported by BMFTR in DAAD project 57616814 (SECAI, School of Embedded Composite AI). MW was partially supported by NSF award CBET-2112085 and DMS-2406905, and an Alfred P. Sloan Research Fellowship in Mathematics.

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

# A. Appendix

## A.1. Extended related work

Numerous works have addressed the challenge of explaining the remarkable success of deep neural networks from diverse theoretical perspectives. One line of research characterizes network expressivity in terms of the complexity of decision boundaries. Pascanu et al. (2013) and Montufar et al. (2014) established bounds on the number of linear regions generated by deep ReLU networks, demonstrating that deep models can generate substantially more linear regions than their shallow counterparts. Furthermore, the Neural Tangent Kernel framework by Jacot et al. (2018) offers an analytical tool to understand the training dynamics of wide networks by relating them to kernel methods.

Other lines of research explore how the geometry and topology of neural feature representations evolve as data propagate through network layers. Using tools from topological data analysis, such as persistent homology, Naitzat et al. (2020) experimentally showed that neural networks progressively simplify the topology of feature representations. Geometric approaches have uncovered similar phenomena of simplification and regularization. Brahma et al. (2015) observed flattening and disentanglement in manifold-shaped data, Ansuini et al. (2019) reported decreasing intrinsic dimension in deeper layers, and Cohen et al. (2020) demonstrated improved classification capacity via geometric simplification.

Beyond empirical observations, several works propose theoretical frameworks building on classical mathematical tools. Hauser & Ray (2017) argued that deep networks can be naturally interpreted using the language of Riemannian geometry, with network layers acting on the coordinate representation of the underlying data manifold. Meanwhile, Haber & Ruthotto (2017) propose to interpret deep learning as a parameter estimation problem for nonlinear dynamical systems, a framework well-suited for analyzing stability and well-posedness of deep learning.

Closest to our work is the framework introduced by Baptista et al. (2024), which evaluates geometric transformations via Ricci flow at a global scale. A comparison between their global analysis and our local analysis is provided in the following section.

### A.1.1. COMPARISON OF LOCAL AND GLOBAL RICCI COEFFICIENTS

Baptista et al. (2024) introduced a metric that quantifies the geometric transformations induced by deep neural networks relative to those predicted by the Ricci flow at a global scale. In this section, we compare their global metric to our local Ricci evolution coefficients.

Their framework is based on comparing the Forman-Ricci curvature at a global scale to a global approximation of the expansion or contraction of the manifold. Specifically, they define

$$\mathcal{F}_\ell = \sum_{e \in E_\ell} \mathcal{F}(e),$$

where $E_\ell$ denotes the edge set of the $k$-nearest-neighbor graph constructed from the set $\Phi_\ell(X) = \{\Phi_\ell(\boldsymbol{x}^{(i)}) : i = 1, \ldots, N\}$. To quantify the global expansion or contraction of the manifold across layers, they consider all pairwise distances:

$$\eta_\ell = \sum_{\boldsymbol{x},\boldsymbol{y} \in \Phi_{\ell+1}(X)} d_{\ell+1}(\boldsymbol{x}, \boldsymbol{y}) - \sum_{\boldsymbol{x},\boldsymbol{y} \in \Phi_\ell(X)} d_\ell(\boldsymbol{x}, \boldsymbol{y}).$$

The relation between these two quantities is then summarized via the Pearson correlation coefficient

$$\rho = \frac{\sum_{\ell=1}^{L-1}(\eta_\ell - \bar{\eta})(\mathcal{F}_\ell - \bar{\mathcal{F}})}{\sqrt{\sum_{\ell=1}^{L-1}(\eta_\ell - \bar{\eta})^2}\sqrt{\sum_{\ell=1}^{L-1}(\mathcal{F}_\ell - \bar{\mathcal{F}})^2}},$$

where $\bar{\eta}$ and $\bar{\mathcal{F}}$ denote the respective layer-wise averages. We will refer to the quantity $\rho$ as the *global Ricci coefficient*. A negative global Ricci coefficient indicates that the geometric changes induced by the network follow the dynamics predicted by Ricci flow at global scale—large "global curvature" corresponds to contraction, while small "global curvature" corresponds to expansion.

Our approach differs in two key aspects. First, it explicitly leverages the inherently local nature of the Ricci flow, which evolves the Riemannian metric tensor at each point of the manifold according to the local curvature, rather than relying on

*Table 2.* Global Ricci coefficients of untrained neural networks, averaged over 100 independently and randomly initialized models.

|  | **Syn-I** | **Syn-II** | **Syn-III** | **Syn-IV** |
|---|---|---|---|---|
| Mean $\pm$ std. | $-0.389 \pm 0.258$ | $-0.349 \pm 0.151$ | $-0.231 \pm 0.193$ | $-0.204 \pm 0.186$ |
| Minimum | $-0.772$ | $-0.644$ | $-0.744$ | $-0.577$ |
| Negative share | 91% | 99% | 89% | 89% |

*Table 3.* Mean local Ricci evolution coefficients of untrained neural networks, averaged over 100 independently and randomly initialized models.

|  | **Syn-I** | **Syn-II** | **Syn-III** | **Syn-IV** |
|---|---|---|---|---|
| Mean $\pm$ std. | $-0.037 \pm 0.077$ | $0.039 \pm 0.052$ | $-0.019 \pm 0.081$ | $-0.035 \pm 0.054$ |
| Minimum | $-0.234$ | $-0.117$ | $-0.173$ | $-0.133$ |
| Negative share | 66% | 21% | 63% | 76% |

global approximations. Second, we adopt the more refined notion of Ollivier-Ricci curvature, which comes with consistency guarantees relative to the curvature of the underlying manifold given sufficiently dense samples (Van Der Hoorn et al., 2021; García Trillos & Weber, 2026). In contrast, Baptista et al. (2024) employ the Forman-Ricci curvature, which cannot capture higher-order structures and is therefore too simplistic to provide a rich geometric characterization.

To propose an early-stopping heuristic, we evaluate the local Ricci evolution coefficients throughout training. The global Ricci coefficient turns out to be too coarse to provide meaningful insights into the learning dynamics. Indeed, even for randomly initialized, untrained networks, the global Ricci coefficient typically takes negative values, suggesting Ricci flow-like behavior. Table 2 reports the global Ricci coefficients of randomly initialized, untrained networks with 10 layers across different datasets, averaged over 100 runs per dataset. For completeness, we also provide the percentage of networks with negative global Ricci coefficient and the minimum observed value.

This phenomenon is consistent with a simple heuristic indicating an inherent negative correlation between $\eta_\ell$ and $\mathcal{F}_\ell$. Specifically, the estimate of the global curvature of the underlying manifold at layer $\ell$ is given by

$$\mathcal{F}_\ell = \sum_{e \in E_\ell} \mathcal{F}(e) = 4|E_\ell| - \sum_{\boldsymbol{x} \in \Phi_\ell(X)} \deg(\boldsymbol{x})^2.$$

From this expression, $\mathcal{F}_\ell$ takes large negative values in densely connected graphs with many high-degree vertices. Such graphs, however, tend to exhibit smaller pairwise distances, thereby yielding larger values of $\eta_\ell$. As a result, a negative correlation between $\eta_\ell$ and $\mathcal{F}_\ell$ is expected regardless of the specific neural network under consideration.

In contrast, when examined using the framework of local Ricci evolution coefficients, no systematic correlation is observed. For randomly initialized networks, the local Ricci evolution coefficients remain close to zero (Table 3), reflecting the lack of correlation between the expansion of local neighborhoods and the Ollivier-Ricci curvature within those neighborhoods. This underscores the value of local Ricci evolution coefficients for studying learning dynamics: since no Ricci flow-like behavior is present at random initialization, they allow us to track the genuine emergence of curvature-driven dynamics during training.

Finally, note that computing the global Ricci coefficient requires the $k$-nearest-neighbor graphs of each layer to be connected. In practice, however, this condition may not be met, especially for smaller values of $k$. In contrast, an advantage of the local Ricci evolution coefficients is that they can still be computed even when the $k$-nearest-neighbor graphs are disconnected. The only requirement is that each point $\boldsymbol{x}$ is connected to its neighbors in the subsequent layer — a significantly weaker condition.

## A.2. Extended background

### A.2.1. APPROXIMATION OF OLLIVIER-RICCI CURVATURE

Computing the Ollivier-Ricci curvature is computationally demanding, since it involves solving an optimal transport problem for every edge in the graph. Using the Hungarian algorithm, each such computation has complexity $O(\deg_{\max}^3)$. However, the computational burden can be alleviated by approximating the Ollivier-Ricci curvature. Tian et al. (2025) proposed an approximation by taking the arithmetic mean of an upper and a lower bound, each of which can be efficiently computed in linear time. These bounds were first established by Jost & Liu (2014).

**Theorem A.1** (Jost & Liu (2014))**.** *Let $\mathcal{G} = (V, E)$ be a locally finite graph and let $u, v \in V$ with $u \sim v$. Then, the Ollivier-Ricci curvature is bounded from below by*

$$\mathcal{O}(u,v) \geq - \left( 1 - \frac{1}{\deg(u)} - \frac{1}{\deg(v)} - \frac{|N(u) \cap N(v)|}{\deg(u) \wedge \deg(v)} \right)_+$$
$$- \left( 1 - \frac{1}{\deg(u)} - \frac{1}{\deg(v)} - \frac{|N(u) \cap N(v)|}{\deg(u) \vee \deg(v)} \right)_+ + \frac{|N(u) \cap N(v)|}{\deg(u) \vee \deg(v)}.$$

*Furthermore, the Ollivier-Ricci curvature is bounded from above by*

$$\mathcal{O}(u,v) \leq \frac{|N(u) \cap N(v)|}{\deg(u) \vee \deg(v)}.$$

Using these bounds, Tian et al. (2025) propose to approximate the Ollivier-Ricci curvature by taking the arithmetic mean, i.e.,

$$\widetilde{\mathcal{O}}(u,v) = \frac{1}{2} \left( \mathcal{O}^{up}(u,v) + \mathcal{O}^{low}(u,v) \right),$$

where $\mathcal{O}^{up}(u,v)$ and $\mathcal{O}^{low}(u,v)$ denote the upper and lower bound established in Theorem A.1. Note that this approximation can be computed with complexity $O(\deg_{\max})$, which strongly reduces the cost compared to computing the exact Ollivier-Ricci curvature.

### A.2.2. FORMAN-RICCI CURVATURE AND ITS AUGMENTATIONS

Forman (2003) introduced a discretization of the classical Ricci curvature on CW complexes, derived from a discrete analogue of the Bochner-Weitzenböck formula. Viewing a simple graph as a one-dimensional CW complex, with edges corresponding to one-cells, allows this notion to be applied naturally to graphs. In particular, for a simple, unweighted graph, the Forman-Ricci curvature of an edge $u \sim v$ is defined as

$$\mathcal{F}(u,v) = 4 - \deg(u) - \deg(v).$$

Although this definition is well-founded in Forman's framework and computationally efficient, it is often too simplistic to provide the rich geometric characterization required in many practical and theoretical applications. For example, a key limitation of the Forman-Ricci curvature is that it disregards the number of triangles adjacent to an edge, one of the most elementary and important geometric properties of a graph Jost & Liu (2014).

To address this limitation, augmentations of the Forman-Ricci curvature have been considered (Bloch, 2014; Samal et al., 2018; Weber et al., 2018). The core idea is to incorporate additional information about the local geometry by constructing a two-dimensional CW-complex from the graph, inserting two-cells into cycles up to a given length. This approach provides a natural way to capture higher-order correlations among vertices in the network. We augment the Forman-Ricci curvature with all cycles of length three, balancing improved empirical performance in community detection (Fesser et al., 2024) with computational tractability. The resulting augmented Forman-Ricci curvature for an edge $u \sim v$ is given by the following combinatorial formula:

$$\mathcal{AF}(u,v) = 4 - \deg(u) - \deg(v) + 3|N(u) \cap N(v)| = \mathcal{F}(u,v) + 3|N(u) \cap N(v)|.$$

This approximation can be computed in $O(E \deg_{\max})$ time on the whole graph, significantly reducing the cost relative to the computation of Ollivier–Ricci curvature.

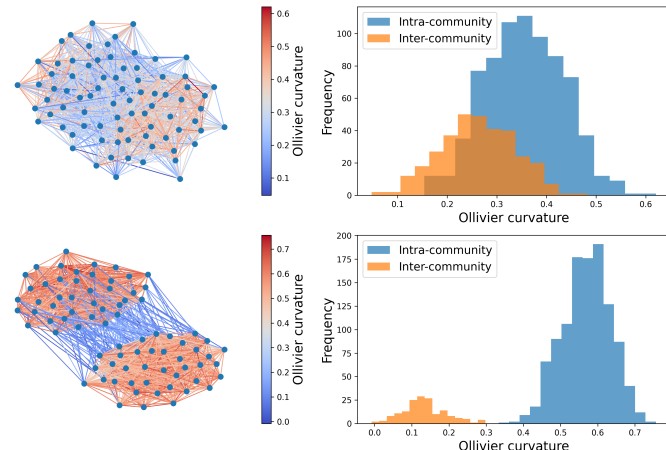

*Figure 7.* Distribution of Ollivier–Ricci curvature for two stochastic block models. The first row shows weak community structure with two communities of 40 nodes each, intra-community edge probability 0.5, and inter-community edge probability 0.2. The second row shows strong community structure with intra-community edge probability 0.7, and inter-community edge probability 0.1.

### A.2.3. MEASURES OF COMMUNITY STRENGTH

Beyond curvature-based measures, the strength of community structure is often assessed using a set of well-established classical metrics. For completeness, we summarize the most widely used ones below. We consider a graph $\mathcal{G} = (V, E)$, where the vertex set is partitioned into disjoint communities $C_1, \ldots, C_n$, i.e.,

$$V = \bigsqcup_{i=1}^{n} C_i.$$

**Modularity.** One of the most prevalent measures for assessing community strength is modularity, first introduced by Newman (2004). It quantifies the density of edges within communities relative to the expected density in a random graph with the same degree distribution. Formally, it is defined by

$$Q = \frac{1}{2|E|} \sum_{u,v \in V} \left( A_{uv} - \frac{\deg(u) \deg(v)}{2|E|} \right) \delta(C_u, C_y),$$

where $\delta(C_u, C_v)$ denotes the Kronecker delta, which equals 1 if $u$ and $v$ belong to the same community and 0 otherwise. Modularity equal to zero indicates that the density of intra-community edges is no greater than what would be expected in a random graph with the same degree distribution. Positive modularity, on the other hand, indicates a higher density of intra-community edges, with values above 0.3 typically reflecting strong community structure.

**Normalized Cut.** Another classical approach for assessing the strength of community structure is based on the cut size, i.e., the number of edges crossing between different communities. Since raw cut size tends to favor unbalanced partitions, Shi & Malik (2000) introduced a normalized variant, defined as

$$\text{Ncut}(C_1, \ldots, C_n) = \frac{1}{2} \sum_{i=1}^{n} \frac{\text{cut}(C_i)}{\text{vol}(C_i)},$$

where $\text{cut}(C_i) = |\{u \sim v : u \in C_i, v \notin C_i\}|$, and $\text{vol}(C_i) = \sum_{v \in C_i} \deg(v)$.

**Algebraic connectivity.** There exists a whole field dedicated to the study of graph Laplacians and their spectra, known as spectral graph theory. The eigenvalues and eigenvectors of the graph Laplacian are closely related to community structure, forming the basis of spectral clustering methods. In particular, the second-smallest eigenvalue of the Laplacian, called the algebraic connectivity, reflects how well connected the graph is: it is greater than zero if and only if the graph is connected, and larger values indicate stronger connectivity. For more details, we refer the reader to the comprehensive book by Chung (1997).

**Curvature Gap.** The neighborhoods of two adjacent vertices tend to be more tightly connected when they belong to the same community. This results in a lower transport cost between their neighborhood distributions and thus higher Ollivier-Ricci curvature. Building on this observation, graphs with community structure exhibit a bimodal distribution of curvature values, reflecting the systematic difference between intra-community and inter-community edges. To quantify this separation, Gosztolai & Arnaudon (2021) introduced the curvature gap:

$$\Delta \mathcal{O} = \frac{1}{\sigma} \left( \mathcal{O}_{\text{intra}} - \mathcal{O}_{\text{inter}} \right),$$

where $\mathcal{O}_{\text{intra}}$ denotes the average curvature of intra-community edges, $\mathcal{O}_{\text{inter}}$ denotes the average curvature of inter-community edges, and $\sigma = \sqrt{\frac{1}{2} \left( \sigma_{\text{intra}}^2 + \sigma_{\text{inter}}^2 \right)}$ is the pooled standard deviation. A large curvature gap indicates a significant distinction in local geometry between edges within communities and those connecting different communities. Figure 7 illustrates this effect for two graphs with different degrees of community strength.

A.2.4. RICCI FLOW

The Ricci flow, introduced by Hamilton (1982), is a second-order nonlinear partial differential equation for the Riemannian metric. Given a smooth Riemannian manifold $M$ with metric $g$, the Ricci flow evolves the metric according to

$$\begin{cases} \dfrac{\partial}{\partial t} g(t) = -2 \operatorname{Ric}(g(t)), \\ \quad g(0) = g, \end{cases} \tag{1}$$

where $\operatorname{Ric}(g(t))$ denotes the Ricci curvature associated with the time-dependent metric $g(t)$. The constant factor $-2$ is conventional; any negative scalar would yield a qualitatively equivalent evolution under an appropriate time reparametrization.

Hamilton proved the short-time existence of solutions to the Ricci flow for arbitrary smooth initial metrics on compact manifolds.

**Theorem A.2** ((Hamilton, 1982), Theorem 4.2). *The Ricci flow introduced in 1 has a solution for a short time on any compact Riemannian manifold with any initial metric at $t = 0$.*

The prove is based on the Nash-Moser implicit function theorem and also ensures the uniqueness of a short-time solution. Furthermore, Hamilton established the long-time existence theorem, which guarantees the existence and uniqueness of a solution as long as the curvature remains bounded.

**Theorem A.3** ((Hamilton, 1982), Theorem 14.1). *The Ricci flow introduced in 1 has a unique solution on a maximal time interval $[0, T)$ with $T \leq \infty$ for any compact Riemannian manifold $M$ with any initial metric at $t = 0$. If $T < \infty$, then*

$$\sup_{x \in M} |\operatorname{Rm}(g(t))|(x) \to \infty$$

*as $t \to T$, where $|\operatorname{Rm}(g(t))|$ denotes the norm of the Riemannian curvature tensor associated with the metric $g(t)$.*

In mathematics, the Ricci flow has obtained significant attention as a tool for proving Thurston's Geometrization Conjecture. This conjecture asserts that every closed 3-manifold can be decomposed in a canonical way into pieces, each admitting one of eight distinct geometric structures, now often called Thurston's model geometries. The Geometrization Conjecture can be viewed as a three-dimensional analogue of the classical Uniformization Theorem for 2-dimensional surfaces, which states that every simply connected Riemannian surface is conformally equivalent to either the Riemann sphere, the complex plane, or the open unit disk.

Hamilton developed a program to prove the Geometrization Conjecture using Ricci flow. While the Ricci flow produces singularities, Hamilton proposed that one might be able to continue the flow past such singularities by using a procedure called "surgery", which cuts the manifold at singular regions and then continues the flow on the resulting pieces. Perelman's breakthrough in 2003 completed this program, introducing a rigorous framework for Ricci flow with surgery and proving the Geometrization Conjecture (Perelman, 2002; 2003b;a). Furthermore, Perelman showed that for simply connected 3-manifolds, the Ricci flow with surgery becomes extinct in finite time. When extinction occurs, Perelman showed that all connected components before extinction must have been round three-dimensional spheres. Using this, he was able to prove the Poincaré conjecture, one of the well-known Millennium Prize Problems.

The topological interpretation of Ricci flow, that it reveals fundamental structure by smoothing geometry, directly parallels what we observe in neural networks. Just as the Ricci flow uncovers the topological type of a manifold by evolving its metric, neural networks reveal the class structure by evolving the geometry of feature representations. The emergence of community structure in our graph representations corresponds to the emergence of distinct geometric pieces in the manifold decomposition.

Interestingly, Ricci flow emerged independently in theoretical physics around the same time, appearing in the work of Friedan (1980). In quantum field theory, Ricci flow arises as a leading-order approximation to the renormalization group flow of the two-dimensional nonlinear $\sigma$-model; see, e.g., Gawedzki (1996-1997). For a comprehensive overview of applications of Ricci flow in physics, we refer the reader to Woolgar (2008).

Since the input manifold in our framework is not directly observed, and the layers of a deep neural network can be viewed as discrete time steps, our focus naturally shifts to discrete formulations of Ricci flow. In passing from the smooth setting to discrete models, one discretizes both the metric and the curvature. On graphs, for example, the Riemannian metric is commonly replaced by edge weights and Ricci curvature is replaced by a discrete curvature $\kappa_t(e)$ attached to each edge $e$. A prototypical discrete Ricci flow then updates the edge weights according to a curvature-driven rule such as

$$w_{t+1}(e) = w_t(e)\big(1 - \eta\,\kappa_t(e)\big),$$

with $\eta > 0$ controlling the step size. Under this update rule, edges with large positive curvature contract while those with negative curvature expand, closely reflecting the behavior of the continuous Ricci flow $\partial_t g = -2\operatorname{Ric}(g)$.

Based on this idea, a variety of discrete Ricci flows, arising from different discretization of Ricci curvature on graphs, have been developed. Ollivier (2010) introduced a discrete Ricci flow using Ollivier-Ricci curvature. Later work established convergence and uniqueness results, such as Li & Münch (2026) for discrete-time flows, and Bai et al. (2024) for continuous-time flows on weighted graphs. Other flows based on different curvature notions include the Bakry-Émery flow (Cushing et al., 2025) and Forman-Ricci flow (Weber et al., 2017). Additionally, Erbar & Kopfer (2020) introduce a concept of super Ricci flow for weighted graphs.

These discrete versions of Ricci flow on graphs have been explored in several machine learning contexts, including applications to community detection (Ni et al., 2019; Sia et al., 2019; Fesser et al., 2024; Gosztolai & Arnaudon, 2021; Tian et al., 2025) and to graph neural networks (Chen et al., 2025).

Another discretization of Ricci flow, the discrete surface Ricci flow, has found many applications in enginneering fields. Here, the discrete surface Ricci flow can be used to design a Riemannian metric, which is conformal to the original metric and induces a user-defined Gaussian-curvature function on the surface (Jin et al., 2008).

Beyond graphs, the discrete surface Ricci flow has become widely adopted in engineering applications. Here, it can be used for designing Riemannian metrics, which are conformal to the original metric and induce a user-specified Gaussian curvature function (Jin et al., 2008). In engineering and computer graphics, surface Ricci flow has been applied to a variety of tasks, including surface parameterization (Jin et al., 2008), 3D shape analysis (Gu et al., 2007b), and the construction of manifold splines (Gu et al., 2007a). For a broader overview of applications across medical imaging, computer graphics, computer vision, and wireless sensor networks, we refer the reader to Zeng & Gu (2013). Notably, the underlying principle parallels our neural network analysis: just as surface Ricci flow deforms metrics to satisfy geometric constraints, we observe that neural feature spaces evolve analogous to Ricci flow to satisfy task-relevant geometric objectives such as class separation in classification.

### A.3. Deferred proof details

In this section, we provide the deferred proofs for the theoretical results stated in Section 3.1.

### A.3.1. RANDOM INITIALIZATION

To derive lower bounds on the network width that ensure the preservation of graph structures under random initialization, we build upon the Johnson–Lindenstrauss Lemma.

**Theorem A.4** (Johnson-Lindenstrauss Lemma, (Johnson et al., 1984)). *Let $\boldsymbol{x} \in \mathbb{R}^n$ and let $\boldsymbol{A} \in \mathbb{R}^{m \times n}$ be a random matrix*

*with i.i.d. entries $A_{ij} \sim \mathcal{N}(0,1)$. Then, for $0 < \epsilon < 1$, we have*

$$\mathbb{P}\left((1-\epsilon)\|\boldsymbol{x}\|^2 \leq \left\|\frac{1}{\sqrt{m}}\boldsymbol{A}\boldsymbol{x}\right\|^2 \leq (1+\epsilon)\|\boldsymbol{x}\|^2\right) \geq 1 - 2\exp\left(\frac{m}{4}(\epsilon^3 - \epsilon^2)\right).$$

*Proof.* Let $\boldsymbol{x} \in \mathbb{R}^n$ be arbitrary. First, observe that the entries of $\boldsymbol{A}\boldsymbol{x}$ are normally distributed, as the sum of independent, normally distributed random variables. Furthermore, we have

$$\mathbb{E}\left[(\boldsymbol{A}\boldsymbol{x})_i\right] = \mathbb{E}\left[\sum_{j=1}^{n} A_{ij}x_j\right] = \sum_{j=1}^{n}\mathbb{E}\left[A_{ij}\right]x_j = 0,$$

and

$$\mathbb{V}[(\boldsymbol{A}\boldsymbol{x})_i] = \mathbb{E}\left[((\boldsymbol{A}\boldsymbol{x})_i)^2\right] - \mathbb{E}\left[((\boldsymbol{A}\boldsymbol{x})_i)\right]^2 = \mathbb{E}\left[\left(\sum_{j=1}^{n} A_{ij}x_j\right)^2\right]$$

$$= \mathbb{E}\left[\sum_{k,j=1}^{n} A_{ij}A_{ik}x_j x_k\right] = \sum_{k,j=1}^{n}\mathbb{E}\left[A_{ij}A_{ik}\right]x_j x_k = \sum_{j=1}^{n} x_j^2 = \|\boldsymbol{x}\|^2.$$

Hence, the random variables

$$X_i = \frac{(\boldsymbol{A}\boldsymbol{x})_i}{\|\boldsymbol{x}\|}$$

are i.i.d. with $X_i \sim \mathcal{N}(0,1)$. Therefore, we obtain

$$\mathbb{P}\left(\left\|\frac{1}{\sqrt{m}}\boldsymbol{A}\boldsymbol{x}\right\|^2 > (1+\epsilon)\|\boldsymbol{x}\|^2\right) = \mathbb{P}\left(\left\|\frac{\boldsymbol{A}\boldsymbol{x}}{\|\boldsymbol{x}\|}\right\|^2 > (1+\epsilon)m\right) = \mathbb{P}\left(\sum_{i=1}^{m} X_i^2 > (1+\epsilon)m\right),$$

where $\sum_{i=1}^{m} X_m^2$ is distributed according to the chi-squared distribution with $m$ degrees of freedom. Using standard concentration inequalities for the chi-squared distribution, we obtain

$$\mathbb{P}\left(\left\|\frac{1}{\sqrt{m}}\boldsymbol{A}\boldsymbol{x}\right\|^2 > (1+\epsilon)\|\boldsymbol{x}\|^2\right) \leq e^{\frac{m}{4}(\epsilon^3 - \epsilon^2)}.$$

Analogously, one can prove that

$$\mathbb{P}\left(\left\|\frac{1}{\sqrt{m}}\boldsymbol{A}\boldsymbol{x}\right\|^2 < (1-\epsilon)\|\boldsymbol{x}\|^2\right) \leq e^{\frac{m}{4}(\epsilon^3 - \epsilon^2)}.$$

This concludes the proof. $\square$

Using Boole's inequality, we immediately obtain the following corollary.

**Corollary A.5.** *Let $X \subset \mathbb{R}^n$ be a finite set, and let $\boldsymbol{A} \in \mathbb{R}^{m \times n}$ be a random matrix with i.i.d. entries $A_{ij} \sim \mathcal{N}(0, 1/m)$. Then, for $0 < \epsilon < 1$, we have*

$$\mathbb{P}\left((1-\epsilon)\|\boldsymbol{x} - \boldsymbol{y}\|^2 \leq \|\boldsymbol{A}\boldsymbol{x} - \boldsymbol{A}\boldsymbol{y}\|^2 \leq (1+\epsilon)\|\boldsymbol{x} - \boldsymbol{y}\|^2 : \forall \boldsymbol{x}, \boldsymbol{y} \in X\right) \geq 1 - \delta,$$

*where*

$$\delta = |X|(|X| - 1)\exp\left(\frac{m}{4}(\epsilon^3 - \epsilon^2)\right).$$

We are now prepared to prove Theorem 3.1.

**Theorem 3.1.** *Let $X \subset \mathbb{R}^n$ be a finite set, and assume there exists $0 < \epsilon < 1$ such that*

$$\min_{\substack{Y \subset X \setminus \{x\} \\ |Y|=k}} \max_{y \in Y} \|x - y\|^2 \leq \frac{1-\epsilon}{1+\epsilon} \min_{\substack{Y \subset X \setminus \{x\} \\ |Y|=k+1}} \max_{y \in Y} \|x - y\|^2 \quad \forall x \in X.$$

*Furthermore, let $A \in \mathbb{R}^{m \times n}$ be a random matrix with i.i.d. entries $A_{ij} \sim \mathcal{N}(0, 1/m)$. Then, the map*

$$\psi : X \to AX := \{Ax : x \in X\}; \quad \psi(x) = Ax$$

*is a graph isomorphism between $\mathcal{G}_k(X)$ and $\mathcal{G}_k(AX)$ with probability bounded from below*

$$\mathbb{P}\left(\mathcal{G}_k(X) \cong \mathcal{G}_k(AX) \text{ under } \psi\right) \geq 1 - |X|(|X|-1)e^{\frac{m}{4}(\epsilon^3 - \epsilon^2)}.$$

*Remark* A.6. To bound the probability of error by $\delta$, i.e.,

$$\mathbb{P}\left(\mathcal{G}_k(X) \ncong \mathcal{G}_k(AX) \text{ under } \psi\right) \leq \delta,$$

we have to choose the width of the network

$$m \geq \frac{4(\log(|X|(|X|-1)) - \log(\delta))}{\epsilon^2 - \epsilon^3}.$$

*Proof.* We first prove that $\psi$ is a graph isomorphism, if

$$(1-\epsilon)\|x - y\|^2 \leq \|Ax - Ay\|^2 \leq (1+\epsilon)\|x - y\|^2 \quad \forall x, y \in X. \tag{2}$$

Let $x, y \in X$ such that $x \sim y$ in $\mathcal{G}_k(X)$. Without loss of generality, we may assume that $y$ is among the $k$-nearest neighbors of $x$. We claim that $Ay$ is among the $k$-nearest neighbors of $Ax$. Assume for contradiction that this is not the case. Hence, there exists a $z \in X$, which is not among the $k$-nearest neighbors of $x$, such that

$$\|Ax - Az\| < \|Ax - Ay\|.$$

This contradicts our assumption, since

$$\|Ax - Ay\|^2 \leq (1+\epsilon)\|x - y\|^2 \leq (1-\epsilon)\|x - z\|^2 \leq \|Ax - Az\|^2,$$

where we applied our assumption on $\epsilon$ to obtain the second inequality. Therefore, our assumption is false, implying that $Ay$ belongs to the $k$-nearest neighbors of $Ax$ and therefore $Ax \sim Ay$ in $\mathcal{G}_k(AX)$.

Conversely, let $Ax \sim Ay$ be an arbitrary edge in $\mathcal{G}_k(AX)$, and assume without loss of generality that $Ay$ is among the $k$-nearest neighbors of $Ax$. It remains to show that $x \sim y$ in $\mathcal{G}_k(X)$. Assume for contradiction that this is not the case. Hence, there exists $z \in X$ among the $k$-nearest neighbors of $x$ such that

$$\|Ax - Az\| > \|Ax - Ay\|.$$

This contradicts our assumption, since

$$\|Ax - Az\|^2 \leq (1+\epsilon)\|x - z\|^2 \leq (1-\epsilon)\|x - y\|^2 \leq \|Ax - Ay\|^2,$$

where we again applied our assumption on $\epsilon$ to obtain the second inequality. Thus, the assumption is contradicted, and $x \sim y$ in $\mathcal{G}_k(X)$ must hold.

This concludes the proof that the map $\psi$ is a graph isomorphism, assuming that condition 2 holds. By Corollary A.5, the probability for this is bounded from below by

$$1 - |X|(|X|-1)\exp\left(\frac{m}{4}(\epsilon^3 - \epsilon^2)\right).$$

This concludes the proof. $\qquad \square$

We can prove a similar result for $r$-neighborhood graphs.

**Theorem A.7.** *Let $X \subset \mathbb{R}^n$ be a finite set, and denote by $N(\boldsymbol{x})$ the one-hop neighborhood of $\boldsymbol{x}$ in $G_r(X)$. Choose $0 < \epsilon < 1$ such that*

$$\epsilon < \min \left\{ \frac{r^2 - \max_{\boldsymbol{y} \in N(\boldsymbol{x})} \|\boldsymbol{x} - \boldsymbol{y}\|^2}{\max_{\boldsymbol{y} \in N(\boldsymbol{x})} \|\boldsymbol{x} - \boldsymbol{y}\|^2}, \frac{\min_{\boldsymbol{y} \notin N(\boldsymbol{x})} \|\boldsymbol{x} - \boldsymbol{y}\|^2 - r^2}{\min_{\boldsymbol{y} \notin N(\boldsymbol{x})} \|\boldsymbol{x} - \boldsymbol{y}\|^2} \right\} \quad \forall \boldsymbol{x} \in X.$$

*Furthermore, let $\boldsymbol{A} \in \mathbb{R}^{m \times n}$ be a random matrix with i.i.d. entries $A_{ij} \sim \mathcal{N}(0, 1/m)$. Then, the map*

$$\psi : X \to \boldsymbol{A}X := \{\boldsymbol{A}\boldsymbol{x} : \boldsymbol{x} \in X\}; \quad \psi(\boldsymbol{x}) = \boldsymbol{A}\boldsymbol{x}$$

*is a graph isomorphism between $G_r(X)$ and $G_r(\boldsymbol{A}X)$ with probability bounded from below by*

$$\mathbb{P}\left(G_r(X) \cong G_r(\boldsymbol{A}X) \text{ under } \psi\right) \geq 1 - |X|(|X| - 1) e^{\frac{m}{4}(\epsilon^3 - \epsilon^2)}.$$

*Proof.* We first prove that $\psi$ is a graph isomorphism, if

$$(1 - \epsilon)\|\boldsymbol{x} - \boldsymbol{y}\|^2 \leq \|\boldsymbol{A}\boldsymbol{x} - \boldsymbol{A}\boldsymbol{y}\|^2 \leq (1 + \epsilon)\|\boldsymbol{x} - \boldsymbol{y}\|^2 \quad \forall \boldsymbol{x}, \boldsymbol{y} \in X. \tag{3}$$

Let $\boldsymbol{x} \sim \boldsymbol{y}$ be an arbitrary edge in $G_r(X)$. Using our assumption and the upper bound on $\epsilon$, we obtain

$$\|\boldsymbol{A}\boldsymbol{x} - \boldsymbol{A}\boldsymbol{y}\|^2 \leq (1 + \epsilon)\|\boldsymbol{x} - \boldsymbol{y}\|^2 < r^2.$$

Therefore, we obtain $\boldsymbol{A}\boldsymbol{x} \sim \boldsymbol{A}\boldsymbol{y}$ in $G_r(\boldsymbol{A}X)$. Analogously, consider an arbitrary edge $\boldsymbol{A}\boldsymbol{x} \sim \boldsymbol{A}\boldsymbol{y}$ in $G_r(\boldsymbol{A}X)$. It remains to show that $\boldsymbol{x} \sim \boldsymbol{y}$ in $G_r(X)$. Assume this is not the case. Hence, $\|\boldsymbol{x} - \boldsymbol{y}\| > r$ and therefore

$$\|\boldsymbol{A}\boldsymbol{x} - \boldsymbol{A}\boldsymbol{y}\|^2 \geq (1 - \epsilon)\|\boldsymbol{x} - \boldsymbol{y}\|^2 > r^2,$$

contradicting $\boldsymbol{A}\boldsymbol{x} \sim \boldsymbol{A}\boldsymbol{y}$. Hence, the assumption leads to a contradiction, and we conclude that $\psi$ is a graph isomorphism, provided that (3) holds. By Corollary A.5, the probability for this is bounded from below by

$$1 - |X|(|X| - 1) \exp\left(\frac{m}{4}(\epsilon^3 - \epsilon^2)\right).$$

This concludes the proof. $\qquad \square$

Thus, for sufficiently wide, randomly initialized one-layer networks without non-linear activation functions, the graph structures are preserved. This result can be naturally extended to multi-layer networks in the following way.

**Theorem A.8.** *Let $X \subset \mathbb{R}^n$ be a finite set, and assume there exists $0 < \epsilon < 1$ such that*

$$\min_{\substack{Y \subset X \setminus \{x\} \\ |Y| = k}} \max_{\boldsymbol{y} \in Y} \|\boldsymbol{x} - \boldsymbol{y}\|^2 \leq \left(\frac{1 - \epsilon}{1 + \epsilon}\right)^L \min_{\substack{Y \subset X \setminus \{x\} \\ |Y| = k+1}} \max_{\boldsymbol{y} \in Y} \|\boldsymbol{x} - \boldsymbol{y}\|^2 \quad \forall \boldsymbol{x} \in X.$$

*Furthermore, let $\boldsymbol{A}_1 \in \mathbb{R}^{m \times n}$ and $\boldsymbol{A}_2, \ldots, \boldsymbol{A}_L \in \mathbb{R}^{m \times m}$ be random matrices with i.i.d. entries $(A_\ell)_{ij} \sim \mathcal{N}(0, 1/m)$ for $\ell = 1, \ldots, L$. Then, the map*

$$\psi_L : X \to X_L := \{\boldsymbol{A}_L \boldsymbol{A}_{L-1} \ldots \boldsymbol{A}_1 \boldsymbol{x} : \boldsymbol{x} \in X\}; \quad \psi_L(\boldsymbol{x}) = \boldsymbol{A}_L \boldsymbol{A}_{L-1} \ldots \boldsymbol{A}_1 \boldsymbol{x}$$

*is a graph isomorphism between $\mathcal{G}_k(X)$ and $\mathcal{G}_k(X_L)$ with probability bounded from below*

$$\mathbb{P}\left(\mathcal{G}_k(X) \cong \mathcal{G}_k(X_L) \text{ under } \psi_L\right) \geq \left(1 - |X|(|X| - 1) e^{\frac{m}{4}(\epsilon^3 - \epsilon^2)}\right)^L.$$

*Proof.* We first prove that $\psi_L$ is an isomorphism, if the following inequality holds for all $x, y \in X$ and $\ell = 1, \ldots, L$:

$$(1 - \epsilon)\|\psi_{\ell-1}(\boldsymbol{x}) - \psi_{\ell-1}(\boldsymbol{y})\|^2 \leq \|\psi_\ell(\boldsymbol{x}) - \psi_\ell(\boldsymbol{y})\|^2 \leq (1 + \epsilon)\|\psi_{\ell-1}(\boldsymbol{x}) - \psi_{\ell-1}(\boldsymbol{y})\|^2, \tag{4}$$

where we use the convention $\psi_0(\boldsymbol{x}) = \boldsymbol{x}$.

To this end, consider an arbitrary edge $\boldsymbol{x} \sim \boldsymbol{y}$ in $\mathcal{G}_k(X)$, and assume without loss of generality that $\boldsymbol{y}$ is among the $k$-nearest neighbors of $\boldsymbol{x}$. We aim to show that $\psi_L(\boldsymbol{x}) \sim \psi_L(\boldsymbol{y})$ in $\mathcal{G}_k(X_L)$. Assume this is not the case. Hence, there exists a vertex $\boldsymbol{z} \in X$, which is not among the $k$-nearest neighbors of $\boldsymbol{x}$, such that

$$\|\psi_L(\boldsymbol{x}) - \psi_L(\boldsymbol{z})\| < \|\psi_L(\boldsymbol{x}) - \psi_L(\boldsymbol{y})\|.$$

This contradicts our assumption, since

$$\|\psi_L(\boldsymbol{x}) - \psi_L(\boldsymbol{y})\|^2 \leq (1+\epsilon)^L \|\boldsymbol{x} - \boldsymbol{y}\|^2 \leq (1-\epsilon)^L \|\boldsymbol{x} - \boldsymbol{z}\|^2 \leq \|\psi_L(\boldsymbol{x}) - \psi_L(\boldsymbol{z})\|^2,$$

where we applied our assumption on $\epsilon$ to obtain the second inequality. Therefore, our assumption is false, implying that $\psi_L(\boldsymbol{x}) \sim \psi_L(\boldsymbol{y})$ in $\mathcal{G}_k(X_L)$ must hold. Using a similar argument, one can show that $\psi_L(\boldsymbol{x}) \sim \psi_L(\boldsymbol{y})$ in $\mathcal{G}_k(X_L)$ implies $\boldsymbol{x} \sim \boldsymbol{y}$ in $\mathcal{G}_k(X)$.

Thus, $\psi_L$ is a graph isomorphism, provided that condition (4) holds. For fixed $\ell$, the probability that this condition is satisfied can be bounded from below by Corollary A.5. Since all entries of all weight matrices are independent, the corresponding events are independent across the different layers $\ell = 1, \ldots, L$. Consequently, we obtain a lower bound by taking the product of the individual probabilities:

$$\mathbb{P}\left(\mathcal{G}_k(X) \cong \mathcal{G}_k(X_L) \text{ under } \phi_L\right) \geq \left(1 - |X|(|X|-1)e^{\frac{m}{4}(\epsilon^3 - \epsilon^2)}\right)^L.$$

$\square$

### A.3.2. TRAINED NETWORKS

Random initialization together with over-parameterization ensures that the network's weights remain close to their initial values throughout gradient descent. To illustrate, consider a two-layer neural network $\Phi = \phi_2 \circ \phi_1$, where the first layer is

$$\phi_1(\boldsymbol{x}) = \sigma\left(\frac{1}{\sqrt{m}}\boldsymbol{W}\boldsymbol{x}\right)$$

with $\sigma$ denoting the ReLU activation and $\boldsymbol{W} \in \mathbb{R}^{m \times n}$ the weight matrix. The second layer computes a weighted linear combination, $\phi_2(\boldsymbol{x}) = \langle \boldsymbol{a}, \boldsymbol{x} \rangle$ with $\boldsymbol{a} \in \mathbb{R}^m$.

Given a training data set $\{(\boldsymbol{x}_i, y_i)\}_{i=1}^N$, we aim to minimize the empirical loss

$$L(\boldsymbol{W}, \boldsymbol{a}) = \frac{1}{2}\sum_{i=1}^N (\Phi(\boldsymbol{x}_i) - y_i)^2.$$

To this end, we fix the second-layer weights $a \in \mathbb{R}^m$ and apply gradient descent to optimize the first-layer weight matrix $\boldsymbol{W} \in \mathbb{R}^{m \times n}$ via the update rule

$$\boldsymbol{W}(k+1) = \boldsymbol{W}(k) - \eta\frac{\partial L(\boldsymbol{W}(k), \boldsymbol{a})}{\partial \boldsymbol{W}(k)},$$

where $\eta > 0$ denotes the learning rate. We denote by

$$\boldsymbol{u}(k) = (\Phi(x_1), \ldots, \Phi(x_N)) \in \mathbb{R}^N$$

the prediction vector after $k$ steps of gradient descent. Our main result in this section relies on an assumption regarding the smallest eigenvalue of the Gram matrix, so we briefly recall this concept here.

**Definition A.9** (Gram matrix). The *Gram matrix* $\boldsymbol{H}^\infty \in \mathbb{R}^{N \times N}$ is defined by

$$H_{ij}^\infty = \mathbb{E}_{\boldsymbol{w} \sim \mathcal{N}(0,1)^n}\left[\boldsymbol{x}_i^\top \boldsymbol{x}_j 1_{\{\boldsymbol{w}^\top \boldsymbol{x}_i \geq 0, \boldsymbol{w}^\top \boldsymbol{x}_j \geq 0\}}\right].$$

We denote by $\lambda_0 = \lambda_{min}(\boldsymbol{H}^\infty)$ the smallest eigenvalue of the Gram matrix.

*Remark* A.10. If $\boldsymbol{x}_i \not\parallel \boldsymbol{x}_j$ for all $i \neq j$, then $\lambda_0 > 0$. Since this condition is typically satisfied in real-world datasets, the assumption $\lambda_0 > 0$ is not restrictive in practice.

Assuming that the smallest eigenvalue of the Gram matrix is strictly positive, Du et al. (2018) proved that gradient descent converges to a global minimum at a linear rate.

**Theorem A.11** ((Du et al., 2018)). *Suppose that $\lambda_0 > 0$ and $\|\boldsymbol{x}_i\| = 1$ and $|y_i| \leq C$ for all $i = 1, \ldots, N$. Assume that the width $m = \Omega\left(\frac{N^6}{\lambda_0^4 \delta^3}\right)$, and $W_{ij} \sim \mathcal{N}(0, 1)$, $a_i \sim Unif(\{-1, 1\})$, and set the step size $\eta = \mathcal{O}\left(\frac{\lambda_0}{N^2}\right)$. Then, with probability at least $1 - \delta$ we obtain*

$$\|\boldsymbol{u}(k) - \boldsymbol{y}\|^2 \leq \left(1 - \frac{\eta \lambda_0}{2}\right)^k \|\boldsymbol{u}(0) - \boldsymbol{y}\|^2.$$

*Remark* A.12. The assumption $\|\boldsymbol{x}_i\| = 1$ can be easily relaxed. If the inputs satisfy $0 < c \leq \|\boldsymbol{x}_i\| \leq C$ for all $i = 1, \ldots, N$, then the result still holds, but the required network width will now also depend on the ratio $\frac{C}{c}$.

Using this, Du et al. (2018) prove that the weight matrix remains close to its initialization throughout training.

**Corollary A.13.** *Assume that the assumptions of Theorem A.11 are satisfied. Then, with probability at least $1 - \delta$, we have for all $k \geq 0$ and every row index $r \in \{1, \ldots, m\}$ that*

$$\|\boldsymbol{W}_{r,:}(k) - \boldsymbol{W}_{r,:}(0)\| \leq \frac{4\sqrt{N}}{\sqrt{m}\lambda_0}\|\boldsymbol{u}(0) - \boldsymbol{y}\|,$$

*where $\boldsymbol{W}_{r,:}(k)$ denotes the $r$-th row of the weight matrix $\boldsymbol{W}(k)$.*

We are now almost ready to prove that, even after an arbitrary number of gradient descent steps, the network remains unable to alter the feature geometry encoded in the graph structures before the application of the nonlinearity. To this end, we introduce one final technical lemma.

**Lemma A.14.** *Let $\boldsymbol{A} \in \mathbb{R}^{m \times n}$ satisfying $\|\boldsymbol{A}_{r,:}\| \leq \epsilon$ for every $r \in \{1, \ldots, m\}$. For $\boldsymbol{x} \in \mathbb{R}^n$, we obtain the following upper bound*

$$\|\boldsymbol{A}\boldsymbol{x}\| \leq \sqrt{m}\epsilon\|\boldsymbol{x}\|.$$

*Proof.* This follows immediately from the Cauchy-Schwarz inequality:

$$\|\boldsymbol{A}\boldsymbol{x}\|^2 = \sum_{r=1}^{m}\langle \boldsymbol{A}_{r,:}, \boldsymbol{x}\rangle^2 \leq \sum_{r=1}^{m}\|\boldsymbol{A}_{r,:}\|^2\|\boldsymbol{x}\|^2 \leq m\epsilon^2\|\boldsymbol{x}\|^2.$$

Taking the square root on both sides completes the proof. $\qquad\square$

We now show that with large probability, sufficiently wide networks cannot alter the geometry of the k-nearest neighbor graph before the activation function is applied, regardless of the number of gradient descent steps performed. This highlights the crucial role of the non-linearity.

**Theorem 3.3.** *Let $X \subset \mathbb{R}^n$ be a finite set, and assume there exists $0 < \epsilon < 1$ such that*

$$\min_{\substack{Y \subset X \setminus \{x\} \\ |Y| = k}} \max_{y \in Y}\|\boldsymbol{x} - \boldsymbol{y}\| \leq \frac{1 - \epsilon}{1 + \epsilon} \min_{\substack{Y \subset X \setminus \{x\} \\ |Y| = k+1}} \max_{y \in Y}\|\boldsymbol{x} - \boldsymbol{y}\| \quad \forall \boldsymbol{x} \in X.$$

*Assume that the assumptions of Theorem A.11 are satisfied. Furthermore, assume that*

$$m \geq \frac{64N\|\boldsymbol{u}(0) - \boldsymbol{y}\|^2}{\epsilon^2 \lambda_0^2}.$$

*Then, for any number of gradient descent steps $l \geq 0$, the map*

$$\psi : X \to X(l) := \left\{\frac{1}{\sqrt{m}}\boldsymbol{W}(l)\boldsymbol{x} : \boldsymbol{x} \in X\right\}; \quad \psi(\boldsymbol{x}) = \frac{1}{\sqrt{m}}\boldsymbol{W}(l)\boldsymbol{x}$$

*is a graph isomorphism between $\mathcal{G}_k(X)$ and $\mathcal{G}_k(X(l))$ with probability bounded from below by*

$$\mathbb{P}(\mathcal{G}_k(X) \cong \mathcal{G}_k(X(l)) \text{ under } \psi) \geq 1 - \delta - |X|(|X| - 1)e^{\frac{m}{4}\left(\frac{\epsilon^3}{8} - \frac{\epsilon^2}{4}\right)}.$$

*Proof.* For ease of notation, we define $\boldsymbol{A}(l) = \frac{1}{\sqrt{m}}\boldsymbol{W}(l)$. Note that the matrix $A(0)$ has i.i.d. entries with $A(0)_{ij} \sim \mathcal{N}(0, 1/m)$. We first show that $\psi$ is a graph isomorphism, if

$$(1 - \frac{\epsilon}{2})\|\boldsymbol{x} - \boldsymbol{y}\|^2 \le \|\boldsymbol{A}(0)(\boldsymbol{x} - \boldsymbol{y})\|^2 \le (1 + \frac{\epsilon}{2})\|\boldsymbol{x} - \boldsymbol{y}\|^2 \quad \forall \boldsymbol{x}, \boldsymbol{y} \in X, \tag{5}$$

and for every $l \ge 0$

$$\|\boldsymbol{W}_{r,:}(l) - \boldsymbol{W}_{r,:}(0)\| \le \frac{4\sqrt{N}}{\sqrt{m}\lambda_0}\|\boldsymbol{u}(0) - \boldsymbol{y}\|. \tag{6}$$

To this end, observe that, for any $\boldsymbol{x} \in \mathbb{R}^n$, we have

$$\|(\boldsymbol{A}(l) - \boldsymbol{A}(0))\boldsymbol{x}\| = \frac{1}{\sqrt{m}}\|(\boldsymbol{W}(l) - \boldsymbol{W}(0))x\| \le \frac{4\sqrt{N}}{\sqrt{m}\lambda_0}\|\boldsymbol{u}(0) - \boldsymbol{y}\|\|\boldsymbol{x}\| \le \frac{\epsilon}{2}\|\boldsymbol{x}\|,$$

where the first inequality is a consequence of Lemma A.14, and the second follows from our assumption on $m$. Using this inequalities, we obtain

$$\begin{aligned}\|\boldsymbol{A}(l)\boldsymbol{x} - \boldsymbol{A}(l)\boldsymbol{y}\| &\le \|(\boldsymbol{A}(l) - \boldsymbol{A}(0))(\boldsymbol{x} - \boldsymbol{y})\| + \|\boldsymbol{A}(0)(\boldsymbol{x} - \boldsymbol{y})\| \\ &\le \frac{\epsilon}{2}\|\boldsymbol{x} - \boldsymbol{y}\| + \sqrt{1 + \frac{\epsilon}{2}}\|\boldsymbol{x} - \boldsymbol{y}\| \\ &\le (1 + \epsilon)\|\boldsymbol{x} - \boldsymbol{y}\| \end{aligned}$$

for all $\boldsymbol{x}, \boldsymbol{y} \in X$. On the other hand, using the reverse triangle inequality, we obtain

$$\begin{aligned}\|(\boldsymbol{A}(l)\boldsymbol{x} - \boldsymbol{A}(l))\boldsymbol{y}\| &\ge \left| \|\boldsymbol{A}(0)(\boldsymbol{x} - \boldsymbol{y})\| - \|(\boldsymbol{A}(l) - \boldsymbol{A}(0))(\boldsymbol{x} - \boldsymbol{y})\| \right| \\ &\ge \sqrt{1 - \frac{\epsilon}{2}}\|\boldsymbol{x} - \boldsymbol{y}\| - \frac{\epsilon}{2}\|\boldsymbol{x} - \boldsymbol{y}\| \\ &\ge (1 - \epsilon)\|\boldsymbol{x} - \boldsymbol{y}\|. \end{aligned}$$

for all $\boldsymbol{x}, \boldsymbol{y} \in X$.

We are now prepared to prove that $\psi$ is a graph isomorphism. To this end, let $\boldsymbol{x} \sim \boldsymbol{y}$ be an arbitrary edge in $\mathcal{G}_k(X)$. Without loss of generality, we may assume that $\boldsymbol{y}$ is among the $k$-nearest neighbors of $\boldsymbol{x}$. We claim that $\psi(\boldsymbol{y})$ is among the $k$-nearest neighbors of $\psi(\boldsymbol{x})$. Assume this is not the case. Hence, there exists a $\boldsymbol{z} \in X$, which is not among the $k$-nearest neighbors of $\boldsymbol{x}$, such that

$$\|\psi(\boldsymbol{x}) - \psi(\boldsymbol{z})\| = \|\boldsymbol{A}(l)\boldsymbol{x} - \boldsymbol{A}(l)\boldsymbol{z}\| < \|\boldsymbol{A}(l)\boldsymbol{x} - \boldsymbol{A}(l)\boldsymbol{y}\| = \|\psi(\boldsymbol{x}) - \psi(\boldsymbol{y})\|.$$

This contradicts

$$\|\boldsymbol{A}(l)\boldsymbol{x} - \boldsymbol{A}(l)\boldsymbol{y}\| \le (1 + \epsilon)\|\boldsymbol{x} - \boldsymbol{y}\| \le (1 - \epsilon)\|\boldsymbol{x} - \boldsymbol{z}\| \le \|\boldsymbol{A}(l)\boldsymbol{x} - \boldsymbol{A}(l)\boldsymbol{z}\|,$$

where we applied our assumption on $\epsilon$ to obtain the second inequality. Therefore, our assumption is false, implying that $\psi(\boldsymbol{y})$ belongs to the $k$-nearest neighbors of $\psi(\boldsymbol{x})$ and therefore $\psi(\boldsymbol{x}) \sim \psi(\boldsymbol{y})$ in $\mathcal{G}_k(X(l))$.

Conversely, let $\psi(\boldsymbol{x}) \sim \psi(\boldsymbol{y})$ be an arbitrary edge in $\mathcal{G}_k(X(l))$, and assume without loss of generality that $\psi(\boldsymbol{y})$ is among the $k$-nearest neighbors of $\psi(\boldsymbol{x})$. It remains to show that $\boldsymbol{x} \sim \boldsymbol{y}$ in $\mathcal{G}_k(X)$. Assume for contradiction that this is not the case. Hence, there exists $\boldsymbol{z} \in X$ among the $k$-nearest neighbors of $\boldsymbol{x}$ such that

$$\|\psi(\boldsymbol{x}) - \psi(\boldsymbol{z})\| = \|\boldsymbol{A}(l)\boldsymbol{x} - \boldsymbol{A}(l)\boldsymbol{z}\| > \|\psi(\boldsymbol{x}) - \psi(\boldsymbol{y})\| = \|\boldsymbol{A}(l)\boldsymbol{x} - \boldsymbol{A}(l)\boldsymbol{y}\|.$$

This contradicts our assumption, since

$$\|\boldsymbol{A}(l)\boldsymbol{x} - \boldsymbol{A}(l)\boldsymbol{z}\| \le (1 + \epsilon)\|\boldsymbol{x} - \boldsymbol{z}\| \le (1 - \epsilon)\|\boldsymbol{x} - \boldsymbol{y}\| \le \|\boldsymbol{A}(l)\boldsymbol{x} - \boldsymbol{A}(l)\boldsymbol{y}\|,$$

where we again applied our assumption on $\epsilon$ to obtain the second inequality. Thus, the assumption is contradicted, and $\boldsymbol{x} \sim \boldsymbol{y}$ in $\mathcal{G}_k(X)$ must hold.

Hence, $\psi$ is a graph isomorphism between $\mathcal{G}_k(X)$ and $\mathcal{G}_k(X(l))$, provided that (5) and (6) hold. According to Corollary A.5, the probability that (5) holds is bounded from below by

$$1 - |X|(|X| - 1)\exp\left(\frac{m}{4}\left(\frac{\epsilon^3}{8} - \frac{\epsilon^2}{4}\right)\right).$$

On the other hand, by Corollary A.13, we know that the probability that (6) holds is bounded from below by $1 - \delta$. The claim now follows from the Bonferroni inequality. $\square$

An analogous result can also be established for $r$-neighborhood graphs.

**Theorem A.15.** *Let $X \subset \mathbb{R}^n$ be a finite set, and denote by $N(\boldsymbol{x})$ the one-hop neighborhood of $\boldsymbol{x}$ in $G_r(X)$. Choose $0 < \epsilon < 1$ such that*

$$\epsilon < \min\left\{\frac{r - \max_{\boldsymbol{y} \in N(\boldsymbol{x})}\|\boldsymbol{x} - \boldsymbol{y}\|}{\max_{\boldsymbol{y} \in N(\boldsymbol{x})}\|\boldsymbol{x} - \boldsymbol{y}\|}, \frac{\min_{\boldsymbol{y} \notin N(\boldsymbol{x})}\|\boldsymbol{x} - \boldsymbol{y}\| - r}{\min_{\boldsymbol{y} \notin N(\boldsymbol{x})}\|\boldsymbol{x} - \boldsymbol{y}\|}\right\} \quad \forall \boldsymbol{x} \in X.$$

*Assume that the assumptions of Theorem A.11 are satisfied. Furthermore, assume that*

$$m \geq \frac{64N\|\boldsymbol{u}(0) - \boldsymbol{y}\|^2}{\epsilon^2 \lambda_0^2}.$$

*Then, for any number of gradient descent steps $l \geq 0$, the map*

$$\psi : X \to X(l) := \left\{\frac{1}{\sqrt{m}}\boldsymbol{W}(l)\boldsymbol{x} : \boldsymbol{x} \in X\right\}; \quad \psi(\boldsymbol{x}) = \frac{1}{\sqrt{m}}\boldsymbol{W}(l)\boldsymbol{x}$$

*is a graph isomorphism between $G_r(X)$ and $G_r(X(l))$ with probability bounded from below by*

$$\mathbb{P}(G_r(X) \cong G_r(X(l)) \text{ under } \psi) \geq 1 - \delta - |X|(|X| - 1)e^{\frac{m}{4}\left(\frac{\epsilon^3}{8} - \frac{\epsilon^2}{4}\right)}.$$

*Proof.* For ease of notation, we define $\boldsymbol{A}(l) = \frac{1}{\sqrt{m}}\boldsymbol{W}(l)$. Note that the matrix $\boldsymbol{A}(0)$ has i.i.d. entries with $\boldsymbol{A}(0)_{ij} \sim \mathcal{N}(0, 1/m)$. We first show that $\psi$ is a graph isomorphism, if

$$(1 - \frac{\epsilon}{2})\|\boldsymbol{x} - \boldsymbol{y}\|^2 \leq \|\boldsymbol{A}(0)(\boldsymbol{x} - \boldsymbol{y})\|^2 \leq (1 + \frac{\epsilon}{2})\|\boldsymbol{x} - \boldsymbol{y}\|^2 \quad \forall \boldsymbol{x}, \boldsymbol{y} \in X, \tag{7}$$

and for every $l \geq 0$

$$\|\boldsymbol{W}_{r,:}(l) - \boldsymbol{W}_{r,:}(0)\| \leq \frac{4\sqrt{N}}{\sqrt{m}\lambda_0}\|\boldsymbol{u}(0) - \boldsymbol{y}\|. \tag{8}$$

By employing the same reasoning as in Theorem 3.3, it follows that

$$(1 - \epsilon)\|\boldsymbol{x} - \boldsymbol{y}\| \leq \|\boldsymbol{A}(l)\boldsymbol{x} - \boldsymbol{A}(l)\boldsymbol{y}\| \leq (1 + \epsilon)\|\boldsymbol{x} - \boldsymbol{y}\|$$

holds for every $\boldsymbol{x}, \boldsymbol{y} \in X$ and $l \geq 0$.

We now proceed to show that $\psi$ is a graph isomorphism. To this end, let $\boldsymbol{x} \sim \boldsymbol{y}$ be an arbitrary edge in $G_r(X)$. Using our upper bound, we obtain

$$\|\psi(\boldsymbol{x}) - \psi(\boldsymbol{y})\| = \|\boldsymbol{A}(l)\boldsymbol{x} - \boldsymbol{A}(l)\boldsymbol{y}\| \leq (1 + \epsilon)\|\boldsymbol{x} - \boldsymbol{y}\| < r,$$

by our assumption on $\epsilon$. Therefore, we conclude $\psi(\boldsymbol{x}) \sim \psi(\boldsymbol{y})$ in $G_r(X(l))$. Conversely, consider an arbitrary edge $\psi(\boldsymbol{x}) \sim \psi(\boldsymbol{y})$ in $G_r(X(l))$. It remains to show that $\boldsymbol{x} \sim \boldsymbol{y}$ in $G_r(X)$. Assume this is not the case. Hence, $\|\boldsymbol{x} - \boldsymbol{y}\| > r$ and therefore

$$\|\psi(\boldsymbol{x}) - \psi(\boldsymbol{y})\| = \|\boldsymbol{A}(l)\boldsymbol{x} - \boldsymbol{A}(l)\boldsymbol{y}\| \geq (1 - \epsilon)\|\boldsymbol{x} - \boldsymbol{y}\| > r,$$

contradicting $\psi(\boldsymbol{x}) \sim \psi(\boldsymbol{y})$.

Hence, $\psi$ is a graph isomorphism between $G_r(X)$ and $G_r(X(l))$, provided that (7) and (8) hold. Again, according to Corollary A.5, the probability that (7) holds is bounded from below by

$$1 - |X|(|X| - 1)\exp\left(\frac{m}{4}\left(\frac{\epsilon^3}{8} - \frac{\epsilon^2}{4}\right)\right).$$

On the other hand, by Corollary A.13, we know that the probability that (8) holds is bounded from below by $1 - \delta$. The claim now follows from the Bonferroni inequality. $\square$

A.3.3. Impact of nonlinearity on feature geometry

We have seen that, in wide linear networks, the feature geometry captured by the graph structures remains unchanged, as the learned weight matrices act approximately as isometries. In this section, we show that this behavior changes once a ReLU activation is introduced. Specifically, we prove that, even when the weight matrices are exact isometries, adding the ReLU nonlinearity is sufficient to change the geometry of the feature manifolds.

It is a standard result from linear algebra that the linear isometries of $\mathbb{R}^n$ correspond precisely to the set of orthogonal matrices, denoted by $\mathrm{O}(n)$, defined as

$$\mathrm{O}(n) = \{\boldsymbol{A} \in \mathbb{R}^{n \times n} : \boldsymbol{A}^\top \boldsymbol{A} = \boldsymbol{I}_n\}.$$

The proof of the main theorem of this section relies on the following lemma.

**Lemma A.16.** *Let $\boldsymbol{x} \in \mathbb{R}^n$ be arbitrary. Then, there exists a linear isometry $\boldsymbol{A} \in \mathrm{O}(n)$ such that*

$$\boldsymbol{A}\boldsymbol{x} = (\|\boldsymbol{x}\|, 0, \ldots, 0)^\top.$$

*Proof.* If $\|\boldsymbol{x}\| = 0$, the claim holds for every linear isometry $\boldsymbol{A} \in \mathrm{O}(n)$. Hence, assume $\|\boldsymbol{x}\| > 0$, and define the normalized vector

$$\boldsymbol{u}_1 = \frac{\boldsymbol{x}}{\|\boldsymbol{x}\|}.$$

Using the Basis Extension Theorem and the Gram-Schmidt Process, we can extend the set $\{\boldsymbol{u}_1\}$ to an orthogonal basis $\{\boldsymbol{u}_1, \ldots, \boldsymbol{u}_n\}$ of $\mathbb{R}^n$. Then, the matrix

$$\boldsymbol{A} = \begin{pmatrix} \boldsymbol{u}_1^\top \\ \vdots \\ \boldsymbol{u}_n^\top \end{pmatrix} \in \mathrm{O}(n)$$

satisfies $\boldsymbol{A}\boldsymbol{x} = (\|\boldsymbol{x}\|, 0, \ldots, 0)^\top$ by construction. $\qquad\square$

We are now prepared to prove the main theorem of this section.

**Theorem 3.4.** *Let $\boldsymbol{x}, \boldsymbol{y}, \boldsymbol{z} \in \mathbb{R}^n$, such that $\boldsymbol{z} \notin \mathrm{span}\{\boldsymbol{x}, \boldsymbol{y}\}$ and*

$$\|\boldsymbol{x} - \boldsymbol{y}\| \geq \|\boldsymbol{x} - \boldsymbol{z}\|.$$

*Then, for $m \geq n$, there exists a linear isometry $\boldsymbol{A} \in \mathbb{R}^{m \times n}$ and a bias vector $\boldsymbol{b} \in \mathbb{R}^m$, such that*

$$\|\sigma(\boldsymbol{A}\boldsymbol{x} + \boldsymbol{b}) - \sigma(\boldsymbol{A}\boldsymbol{y} + \boldsymbol{b})\| < \|\sigma(\boldsymbol{A}\boldsymbol{x} + \boldsymbol{b}) - \sigma(\boldsymbol{A}\boldsymbol{z} + \boldsymbol{b})\|.$$

*Remark A.17.* As shown above, a wide linear neural network cannot change the geometry of the features, since its weight matrices are almost isometries. However, as Theorem 3.4 demonstrates, this is no longer the case once the ReLU activation function is introduced: for any three vertices, the ordering of their pairwise distances can be altered by applying an orthogonal matrix followed by the ReLU activation, thereby rewiring the $k$-nearest neighbor graph.

*Proof.* Without loss of generality, we may assume $m = n$. In the case $m > n$, any $n$-dimensional vector can be embedded into $\mathbb{R}^m$ by appending $m - n$ zero coordinates. If $\|\boldsymbol{x}\| = 0$, then by assumption $\|\boldsymbol{y}\| > 0$ must hold. According to Lemma A.16, there exists $\boldsymbol{A}_1 \in \mathrm{O}(n)$, such that

$$\boldsymbol{A}_1\boldsymbol{y} = (-\|\boldsymbol{y}\|, 0, \ldots, 0)^\top.$$

By assumption, we have $\boldsymbol{z} \notin \mathrm{span}\{\boldsymbol{x}, \boldsymbol{y}\} = \mathrm{span}\{\boldsymbol{y}\}$. Therefore, there exists $i \in \{2, \ldots, n\}$ such that $(\boldsymbol{A}_1\boldsymbol{z})_i \neq 0$. Without loss of generality, we may assume that $(\boldsymbol{A}_1\boldsymbol{z})_i > 0$. Choose $\boldsymbol{b}$ to be the zero vector in $\mathbb{R}^n$. Then,

$$\|\sigma(\boldsymbol{A}_1\boldsymbol{x} + \boldsymbol{b}) - \sigma(\boldsymbol{A}_1\boldsymbol{y} + \boldsymbol{b})\| = 0 < ((\boldsymbol{A}_1\boldsymbol{z})_i)^2 \leq \|\sigma(\boldsymbol{A}_1\boldsymbol{x} + \boldsymbol{b}) - \sigma(\boldsymbol{A}_1\boldsymbol{z} + \boldsymbol{b})\|.$$

Thus, we may assume $\|x\| > 0$.

According to Lemma A.16, there exists $\boldsymbol{A}_1 \in O(n)$, such that

$$\boldsymbol{A}_1 \boldsymbol{x} = (-\|\boldsymbol{x}\|, 0, \ldots, 0)^\top.$$

The proof proceeds by cases.

*Case 1:* $\boldsymbol{y} \in \text{span}\{\boldsymbol{x}\}$. Hence, there exists $\alpha \in \mathbb{R}$ such that $\boldsymbol{y} = \alpha \boldsymbol{x}$. Thus, we obtain

$$\boldsymbol{A}_1 \boldsymbol{y} = (-\alpha\|\boldsymbol{x}\|, 0, \ldots, 0).$$

By assumption, we have $\boldsymbol{z} \notin \text{span}\{\boldsymbol{x}\}$. Therefore, there exists $i \in \{2, \ldots, n\}$ such that $(\boldsymbol{A}_1 \boldsymbol{z})_i \neq 0$. Without loss of generality, we may assume that $(\boldsymbol{A}_1 \boldsymbol{z})_i > 0$. Define the bias vector

$$\boldsymbol{b} = \begin{cases} (\alpha\|\boldsymbol{x}\|, 0, \ldots, 0), & \text{if } \alpha < 0, \\ \boldsymbol{0}, & \text{otherwise,} \end{cases}$$

where $\boldsymbol{0} \in \mathbb{R}^n$ denotes the zero vector. Thus, by construction, we obtain

$$\|\sigma(\boldsymbol{A}_1 \boldsymbol{x} + \boldsymbol{b}) - \sigma(\boldsymbol{A}_1 \boldsymbol{y} + \boldsymbol{b})\| = 0 < ((\boldsymbol{A}_1 \boldsymbol{z})_i)^2 \leq \|\sigma(\boldsymbol{A}_1 \boldsymbol{x} + \boldsymbol{b}) - \sigma(\boldsymbol{A}_1 \boldsymbol{z} + \boldsymbol{b})\|.$$

*Case 2:* $\boldsymbol{y} \notin \text{span}\{\boldsymbol{x}\}$. Denote by

$$(\boldsymbol{A}_1 \boldsymbol{y})_{-1} = ((\boldsymbol{A}_1 \boldsymbol{y})_2, \ldots, (\boldsymbol{A}_1 \boldsymbol{y})_n) \in \mathbb{R}^{n-1}$$

the vector obtained from $\boldsymbol{A}_1 \boldsymbol{y}$ by removing its first coordinate. Note that $\|(\boldsymbol{A}_1 \boldsymbol{y})_{-1}\| > 0$, since $\boldsymbol{y} \notin \text{span}\{\boldsymbol{x}\}$. Define

$$\tilde{\boldsymbol{u}}_1 = \frac{(\boldsymbol{A}_1 \boldsymbol{y})_{-1}}{\|(\boldsymbol{A}_1 \boldsymbol{y})_{-1}\|}.$$

and $\boldsymbol{u}_1 = (0, -\tilde{\boldsymbol{u}}_1) \in \mathbb{R}^n$, so that the first coordinate of $\boldsymbol{u}_1$ is zero and the remaining coordinates are given by $\tilde{\boldsymbol{u}}_1$.

Denote by $\boldsymbol{e}^{(i)}$ the $i$-th standard basis vector. The set $\{\boldsymbol{e}^{(1)}, \boldsymbol{u}_1\}$ forms an orthonormal system and can therefore be extended to an orthonormal basis $\{\boldsymbol{e}^{(1)}, \boldsymbol{u}_1, \ldots, \boldsymbol{u}_{n-1}\}$ of $\mathbb{R}^n$ using the Basis Extension Theorem together with the Gram–Schmidt Process. Note that for every $i \in \{1, \ldots, n-1\}$, we have $\langle \boldsymbol{u}_i, \boldsymbol{e}^{(1)} \rangle = 0$, and therefore $\langle \boldsymbol{u}_i, \boldsymbol{A}_1 \boldsymbol{x} \rangle = 0$. Define the matrix

$$\boldsymbol{A}_2 = \begin{pmatrix} \boldsymbol{e}^{(1)\top} \\ \boldsymbol{u}_1^\top \\ \vdots \\ \boldsymbol{u}_{n-1}^\top \end{pmatrix} \in O(n) \quad \text{and} \quad \boldsymbol{A} = \boldsymbol{A}_2 \boldsymbol{A}_1 \in O(n).$$

By construction, we obtain

$$\boldsymbol{A}\boldsymbol{x} = (-\|\boldsymbol{x}\|, 0, \ldots, 0)^\top \quad \text{and} \quad \boldsymbol{A}\boldsymbol{y} = ((\boldsymbol{A}_1 \boldsymbol{y})_1, -\|(\boldsymbol{A}_1 \boldsymbol{y})_{-1}\|, 0, \ldots, 0).$$

By assumption, we have $\boldsymbol{z} \notin \text{span}\{\boldsymbol{x}, \boldsymbol{y}\}$. Hence, there exists $i \in \{3, \ldots, n\}$ such that $(\boldsymbol{A}\boldsymbol{z})_i \neq 0$. Without loss of generality, we may assume that $(\boldsymbol{A}\boldsymbol{z})_i > 0$.

Finally, define the bias vector

$$\boldsymbol{b} = \begin{cases} (-(\boldsymbol{A}_1 \boldsymbol{y})_1, 0, \ldots, 0), & \text{if } (\boldsymbol{A}_1 \boldsymbol{y})_1 > 0, \\ \boldsymbol{0}, & \text{otherwise.} \end{cases}$$

Therefore, by construction, we obtain

$$\|\sigma(\boldsymbol{A}\boldsymbol{x} + \boldsymbol{b}) - \sigma(\boldsymbol{A}\boldsymbol{y} + \boldsymbol{b})\| = 0 < ((\boldsymbol{A}\boldsymbol{z})_i)^2 \leq \|\sigma(\boldsymbol{A}\boldsymbol{x} + \boldsymbol{b}) - \sigma(\boldsymbol{A}\boldsymbol{z} + \boldsymbol{b})\|.$$

$\square$

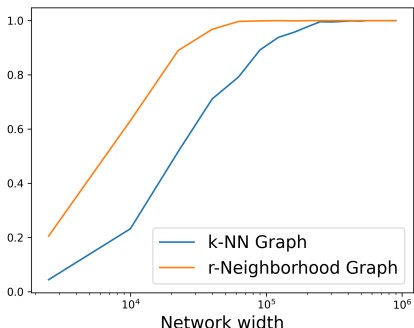

*Figure 8.* Proportion of linear neural networks that preserve the $k$-nearest neighbor and $r$-neighborhood graphs, constructed from the feature manifolds, across different network widths. The graphs are built from a point cloud of 50 samples in the 3-dimensional unit ball. We consider the 5-nearest neighbor graph, and for the $r$-neighborhood graph we set the radius equal to 0.3.

### A.4. Additional experimental results

#### A.4.1. EXPERIMENTAL CONFIRMATION OF THEORETICAL INSIGHTS

We supplement the theoretical results of Section 3.1 with experimental validation. Specifically, we sample points uniformly from the $d$-dimensional unit ball and construct $k$-nearest neighbor graphs on these point clouds. Note that any point cloud can be rescaled to the unit ball without altering its $k$-nearest neighbor graph, ensuring generality of this setup. For varying network widths, we apply randomly initialized neural networks and test whether the induced graphs remain isomorphic to the original ones. Figure 8 reports the proportion of linear neural networks that preserve the $k$-nearest neighbor and $r$-neighborhood graphs across different widths. For each width, 1,000 linear neural networks were independently initialized. Consistent with our theoretical predictions, the preservation probability converges to one as the width increases. The faster convergence observed for $r$-neighborhood graphs is explained by the fact that the maximal $\epsilon$ satisfying the condition of Theorem A.7 was larger than the corresponding bound from Theorem 3.1.

Across all experiments, we find that the network widths required for the estimated probabilities to exceed a given threshold $1 - \delta$ are in practice smaller than the widths for which Theorems 3.1 and A.7 guarantee this. This is expected, since the proofs rely on Boole's inequality, which generally does not provide a tight bound for the probability of a union.

#### A.4.2. LOCAL RICCI EVOLUTION COEFFICIENTS

In this subsection, we present additional experimental results for the computation of local Ricci evolution coefficients. In addition to the Ollivier-Ricci curvature, we also compute the coefficients using the augmented Forman curvature and the approximation of Ollivier-Ricci curvature proposed by Tian et al. (2025). For all curvature notions, we evaluate both our synthetic and real-world datasets by training deep neural networks of varying width and depth and subsequently computing the local Ricci evolution coefficients. We average our results over 50 independently trained networks for each dataset-architecture pair, to account for the inherent randomness in neural network training, making sure our observed patterns are robust rather than accidental.

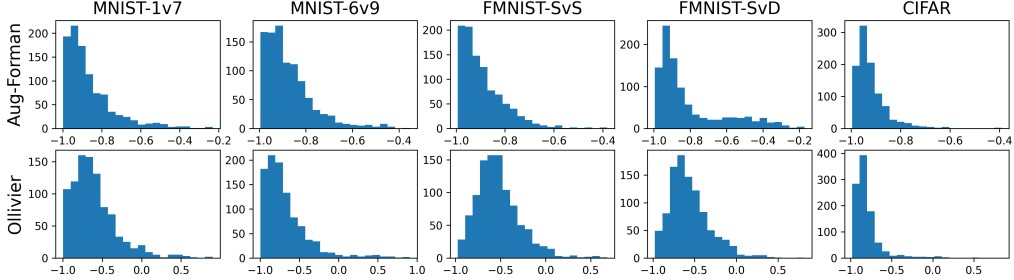

*Figure 9.* Distribution of local Ricci evolution coefficients for networks of depth 15 and width 50 on real-world datasets, shown for augmented Forman–Ricci curvature (top row) and Ollivier–Ricci curvature (bottom row).

*Table 4.* Average local Ricci evolution coefficients, computed using augmented Forman curvature, on real-world data. Values are means $\pm$ standard deviations over 50 independently trained networks per architecture; proportion of vertices with negative coefficients is reported in parentheses. Networks were randomly initialized.

| (Width,Depth) | MNIST-1v7 | MNIST-6v9 | FMNIST-SvS | FMNIST-SvD | CIFAR |
|---|---|---|---|---|---|
| (15, 7) | $-0.82 \pm 0.06\,(98.5\%)$ | $-0.79 \pm 0.09\,(97.5\%)$ | $-0.81 \pm 0.08\,(98.7\%)$ | $-0.53 \pm 0.13\,(91.2\%)$ | $-0.73 \pm 0.11\,(98.2\%)$ |
| (15, 10) | $-0.83 \pm 0.05\,(99.3\%)$ | $-0.83 \pm 0.06\,(99.5\%)$ | $-0.82 \pm 0.06\,(99.8\%)$ | $-0.58 \pm 0.16\,(94.0\%)$ | $-0.76 \pm 0.15\,(97.5\%)$ |
| (15, 15) | $-0.84 \pm 0.05\,(99.8\%)$ | $-0.88 \pm 0.03\,(99.9\%)$ | $-0.86 \pm 0.05\,(99.9\%)$ | $-0.66 \pm 0.13\,(97.5\%)$ | $-0.79 \pm 0.21\,(97.4\%)$ |
| (25, 7) | $-0.80 \pm 0.05\,(97.8\%)$ | $-0.69 \pm 0.18\,(94.0\%)$ | $-0.80 \pm 0.04\,(99.5\%)$ | $-0.54 \pm 0.11\,(91.9\%)$ | $-0.69 \pm 0.13\,(96.7\%)$ |
| (25, 10) | $-0.83 \pm 0.07\,(99.0\%)$ | $-0.83 \pm 0.06\,(99.4\%)$ | $-0.83 \pm 0.05\,(99.8\%)$ | $-0.62 \pm 0.12\,(96.2\%)$ | $-0.80 \pm 0.09\,(99.3\%)$ |
| (25, 15) | $-0.83 \pm 0.05\,(99.8\%)$ | $-0.85 \pm 0.04\,(99.9\%)$ | $-0.85 \pm 0.04\,(100\%)$ | $-0.74 \pm 0.07\,(99.4\%)$ | $-0.90 \pm 0.03\,(99.9\%)$ |
| (50, 7) | $-0.81 \pm 0.04\,(98.3\%)$ | $-0.61 \pm 0.16\,(91.1\%)$ | $-0.79 \pm 0.04\,(99.3\%)$ | $-0.58 \pm 0.10\,(92.9\%)$ | $-0.76 \pm 0.08\,(98.8\%)$ |
| (50, 10) | $-0.84 \pm 0.03\,(99.7\%)$ | $-0.79 \pm 0.09\,(98.5\%)$ | $-0.84 \pm 0.05\,(99.8\%)$ | $-0.70 \pm 0.12\,(97.2\%)$ | $-0.87 \pm 0.03\,(100\%)$ |
| (50, 15) | $-0.83 \pm 0.04\,(99.9\%)$ | $-0.86 \pm 0.05\,(99.9\%)$ | $-0.88 \pm 0.02\,(100\%)$ | $-0.80 \pm 0.06\,(100\%)$ | $-0.91 \pm 0.02\,(100\%)$ |

*Table 5.* Average local Ricci evolution coefficients, computed using approximated Ollivier curvature, on real-world data. Values are means $\pm$ standard deviations over 50 independently trained networks per architecture; proportion of vertices with negative coefficients is reported in parentheses. Networks were randomly initialized.

| (Width,Depth) | MNIST-1v7 | MNIST-6v9 | FMNIST-SvS | FMNIST-SvD | CIFAR |
|---|---|---|---|---|---|
| (15, 7) | $-0.75 \pm 0.21\,(94.6\%)$ | $-0.76 \pm 0.12\,(96.4\%)$ | $-0.77 \pm 0.07\,(98.8\%)$ | $-0.50 \pm 0.13\,(89.5\%)$ | $-0.66 \pm 0.13\,(96.5\%)$ |
| (15, 10) | $-0.82 \pm 0.05\,(99.5\%)$ | $-0.84 \pm 0.04\,(99.7\%)$ | $-0.77 \pm 0.06\,(99.7\%)$ | $-0.54 \pm 0.16\,(92.7\%)$ | $-0.69 \pm 0.16\,(96.4\%)$ |
| (15, 15) | $-0.83 \pm 0.04\,(99.8\%)$ | $-0.83 \pm 0.07\,(99.1\%)$ | $-0.81 \pm 0.06\,(99.9\%)$ | $-0.64 \pm 0.14\,(97.0\%)$ | $-0.75 \pm 0.18\,(97.2\%)$ |
| (25, 7) | $-0.79 \pm 0.05\,(98.1\%)$ | $-0.66 \pm 0.20\,(92.3\%)$ | $-0.76 \pm 0.04\,(99.2\%)$ | $-0.50 \pm 0.12\,(90.0\%)$ | $-0.62 \pm 0.15\,(94.6\%)$ |
| (25, 10) | $-0.81 \pm 0.04\,(99.4\%)$ | $-0.82 \pm 0.05\,(99.6\%)$ | $-0.78 \pm 0.04\,(99.6\%)$ | $-0.59 \pm 0.12\,(95.3\%)$ | $-0.74 \pm 0.10\,(98.9\%)$ |
| (25, 15) | $-0.81 \pm 0.06\,(99.8\%)$ | $-0.84 \pm 0.03\,(100\%)$ | $-0.79 \pm 0.04\,(100\%)$ | $-0.72 \pm 0.08\,(99.2\%)$ | $-0.86 \pm 0.04\,(99.9\%)$ |
| (50, 7) | $-0.79 \pm 0.04\,(98.5\%)$ | $-0.57 \pm 0.21\,(87.9\%)$ | $-0.75 \pm 0.04\,(98.9\%)$ | $-0.56 \pm 0.11\,(91.4\%)$ | $-0.69 \pm 0.10\,(98.1)\%$ |
| (50, 10) | $-0.82 \pm 0.03\,(99.8\%)$ | $-0.82 \pm 0.04\,(99.7\%)$ | $-0.80 \pm 0.05\,(99.7\%)$ | $-0.68 \pm 0.12\,(96.6\%)$ | $-0.83 \pm 0.03\,(100\%)$ |
| (50, 15) | $-0.83 \pm 0.04\,(99.9\%)$ | $-0.84 \pm 0.04\,(100\%)$ | $-0.83 \pm 0.03\,(100\%)$ | $-0.78 \pm 0.07\,(100\%)$ | $-0.89 \pm 0.03\,(100\%)$ |

Results on real-world datasets using augmented Forman curvature and approximated Ollivier curvature are reported in Table 4 and Table 5, respectively. In all cases, we observe strongly negative local Ricci evolution coefficients, highlighting pronounced curvature-driven dynamics in the evolution of feature geometry. To further support this finding, we evaluate the proportion of vertices with negative local coefficients, consistently showing that the vast majority of vertices exhibit such behavior. Hence, curvature-driven dynamics appear almost universally across the data manifold. Figure 9 shows the entire distribution of local Ricci evolution coefficients on the real-world datasets for both Ollivier–Ricci curvature and augmented Forman-Ricci curvature. Complementary results on synthetic datasets are provided in Table 6 for Ollivier-Ricci curvature, in Table 7 for augmented Forman-Ricci curvature, and in Table 8 for approximated Ollivier-Ricci curvature. The results are consistent with those observed on the real-world datasets.

Furthermore, we observe consistent results for all three discretizations of Ricci curvature. The numerical values obtained using the augmented Forman-Ricci curvature and the approximation of the Ollivier-Ricci curvature are nearly identical, which is expected since both curvature notions are primarily influenced by three-cycles. Moreover, Jost & Münch (2021) show that Ollivier–Ricci curvature coincides with the maximal Forman curvature over cell complexes having the given graph as their 1-skeleton, providing a theoretical explanation for the close agreement observed across the different notions. In contrast, we also calculated the local Ricci evolution coefficients using the classical Forman-Ricci curvature, as shown in Table 9. As anticipated, the correlation observed in previous experiments is substantially reduced or absent when using the unaugmented curvature, validating the necessity of a more expressive measure for our purposes.

Above, we computed the Pearson correlation between the Ricci curvature and the local expansion or contraction. As a further analysis, we evaluated the Spearman rank correlation coefficients for the same quantities. This non-parametric measure captures monotonic relationships and is less sensitive to outliers or non-normal distributions. The results presented in Table 10 confirm that the observed correlations are consistent across both Pearson and Spearman metrics.

### A.4.3. OVERFITTING UNDER LABEL NOISE

The analysis in Section 4.3 suggests that local Ricci evolution coefficients can serve as an indicator for the onset of overfitting. To test this interpretation in a controlled setting where overfitting is induced by design, we conducted additional experiments on the synthetic dataset Syn-I in which 25% of the training labels were flipped. This corruption creates a setting in which the network can continue to fit training samples beyond the point at which it has learned the true decision boundary, leading to a clear divergence between training and test accuracy.

*Table 6.* Average local Ricci evolution coefficients, computed using Ollivier curvature, on synthetic data. Values are means $\pm$ standard deviations over 50 independently trained networks per architecture; proportion of vertices with negative coefficients is reported in parentheses. Networks were randomly initialized.

| (Width,Depth) | Syn-I | Syn-II | Syn-III | Syn-IV |
|---|---|---|---|---|
| (15,7) | $-0.38 \pm 0.07\,(80.9\%)$ | $-0.31 \pm 0.11\,(78.2\%)$ | $-0.53 \pm 0.09\,(92.1\%)$ | $-0.39 \pm 0.09\,(82.9\%)$ |
| (15,10) | $-0.43 \pm 0.07\,(83.9\%)$ | $-0.29 \pm 0.14\,(81.2\%)$ | $-0.59 \pm 0.09\,(92.7\%)$ | $-0.45 \pm 0.10\,(87.1\%)$ |
| (15,15) | $-0.43 \pm 0.12\,(81.0\%)$ | $-0.36 \pm 0.09\,(84.9\%)$ | $-0.64 \pm 0.07\,(93.9\%)$ | $-0.49 \pm 0.13\,(86.2\%)$ |
| (25,7) | $-0.37 \pm 0.06\,(81.9\%)$ | $-0.34 \pm 0.10\,(79.7\%)$ | $-0.56 \pm 0.07\,(93.8\%)$ | $-0.32 \pm 0.09\,(77.4\%)$ |
| (25,10) | $-0.43 \pm 0.07\,(83.8\%)$ | $-0.37 \pm 0.08\,(86.9\%)$ | $-0.63 \pm 0.05\,(96.2\%)$ | $-0.40 \pm 0.09\,(85.4\%)$ |
| (25,15) | $-0.49 \pm 0.04\,(86.5\%)$ | $-0.40 \pm 0.05\,(87.6\%)$ | $-0.69 \pm 0.04\,(95.5\%)$ | $-0.51 \pm 0.05\,(90.3\%)$ |
| (50,7) | $-0.38 \pm 0.06\,(83.2\%)$ | $-0.38 \pm 0.07\,(81.6\%)$ | $-0.59 \pm 0.05\,(96.3\%)$ | $-0.29 \pm 0.05\,(74.9\%)$ |
| (50,10) | $-0.47 \pm 0.05\,(88.0\%)$ | $-0.41 \pm 0.05\,(86.9\%)$ | $-0.66 \pm 0.05\,(97.3\%)$ | $-0.34 \pm 0.07\,(81.8\%)$ |
| (50,15) | $-0.53 \pm 0.04\,(89.1\%)$ | $-0.42 \pm 0.04\,(88.2\%)$ | $-0.72 \pm 0.03\,(97.0\%)$ | $-0.53 \pm 0.06\,(91.7\%)$ |

*Table 7.* Average local Ricci evolution coefficients, computed using augmented Forman curvature, on synthetic data. Values are means $\pm$ standard deviations over 50 independently trained networks per architecture; proportion of vertices with negative coefficients is reported in parentheses. Networks were randomly initialized.

| (Width,Depth) | Syn-I | Syn-II | Syn-III | Syn-IV |
|---|---|---|---|---|
| (15,7) | $-0.43 \pm 0.10\,(87.2\%)$ | $-0.32 \pm 0.16\,(78.4\%)$ | $-0.64 \pm 0.08\,(97.6\%)$ | $-0.37 \pm 0.12\,(81.7\%)$ |
| (15,10) | $-0.51 \pm 0.16\,(90.4\%)$ | $-0.34 \pm 0.12\,(87.4\%)$ | $-0.72 \pm 0.10\,(98.5\%)$ | $-0.48 \pm 0.13\,(90.0\%)$ |
| (15,15) | $-0.63 \pm 0.10\,(95.3\%)$ | $-0.45 \pm 0.09\,(91.6\%)$ | $-0.70 \pm 0.15\,(96.0\%)$ | $-0.63 \pm 0.20\,(94.3\%)$ |
| (25,7) | $-0.43 \pm 0.09\,(88.4\%)$ | $-0.36 \pm 0.14\,(81.6\%)$ | $-0.63 \pm 0.08\,(97.6\%)$ | $-0.27 \pm 0.09\,(73.7\%)$ |
| (25,10) | $-0.57 \pm 0.07\,(95.5\%)$ | $-0.40 \pm 0.10\,(91.5\%)$ | $-0.74 \pm 0.06\,(99.3\%)$ | $-0.44 \pm 0.11\,(88.1\%)$ |
| (25,15) | $-0.65 \pm 0.09\,(97.0\%)$ | $-0.50 \pm 0.07\,(95.9\%)$ | $-0.75 \pm 0.12\,(98.2\%)$ | $-0.67 \pm 0.07\,(97.4\%)$ |
| (50,7) | $-0.39 \pm 0.09\,(86.0\%)$ | $-0.45 \pm 0.08\,(89.0\%)$ | $-0.62 \pm 0.07\,(97.4\%)$ | $-0.22 \pm 0.08\,(68.7\%)$ |
| (50,10) | $-0.58 \pm 0.07\,(96.7\%)$ | $-0.50 \pm 0.08\,(95.5\%)$ | $-0.76 \pm 0.05\,(99.8\%)$ | $-0.33 \pm 0.10\,(79.8\%)$ |
| (50,15) | $-0.69 \pm 0.06\,(98.4\%)$ | $-0.55 \pm 0.05\,(97.1\%)$ | $-0.81 \pm 0.03\,(99.8\%)$ | $-0.63 \pm 0.10\,(96.5\%)$ |

We observe behavior consistent with our interpretation. During the initial phase of training, as the network learns the underlying decision boundary, the local Ricci evolution coefficients exhibit a sharp decline. This coincides with test accuracy rising to approximately 95%. Beyond this point, training accuracy continues to improve as the network begins to fit the corrupted labels, while test accuracy drops, a clear signature of overfitting. During this second phase, the Ricci evolution coefficients increase again, indicating that the network's geometric transformations are no longer as strongly aligned with curvature-driven dynamics. This provides further evidence that monitoring local Ricci evolution coefficients during training could serve as a principled stopping heuristic, as the reversal in their trend coincides with the transition from genuine generalization to memorization of noise.

### A.4.4. COMMUNITY STRUCTURE

In this section, we examine how the curvature gap evolves as the data manifold propagates through the layers of the deep neural network. Whereas both modularity and the normalized cut provide clear evidence that the network rewires the $k$-NN graph derived from the point clouds such that its geometry aligns more closely with the community structure induced by the true labels (see Figure 2 and Figure 10), the behavior of the curvature gap is less straightforward.

The explanation for this is that most inter-community edges connect misclassified nodes to correctly classified nodes with the same label, making them indistinguishable from intra-community edges. This effect is clearly illustrated in Figure 11, where we show the full curvature distribution on the MNIST 1-vs-7 dataset, comparing inter-community edges (orange) and intra-community edges (blue). As expected, intra-community edges systematically shift toward more positive curvature values as the $k$-nearest neighbor graphs are transformed through the layers of the deep neural network. In contrast, the behavior of inter-community edges is more intricate. The left column displays the distributions computed on the entire test set. In the final layer, two structurally distinct types of inter-community edges emerge. The majority exhibit positive curvature and vanish once the five misclassified points are removed. These are precisely the edges described above, connecting a misclassified point with a correctly classified one. In contrast, a small subset of inter-community edges remains, characterized by highly negative curvature values. These correspond to the true inter-community edges. This distinction explains the vanishing of the curvature gaps before removing the misclassified samples, and we find the same qualitative pattern consistently across all synthetic and real-world datasets considered.

*Table 8.* Average local Ricci evolution coefficients, computed using approximated Ollivier curvature, on synthetic data. Values are means $\pm$ standard deviations over 50 independently trained networks per architecture; proportion of vertices with negative coefficients is reported in parentheses. Networks were randomly initialized.

| (Width,Depth) | Syn-I | Syn-II | Syn-III | Syn-IV |
|---|---|---|---|---|
| (15,7) | $-0.44 \pm 0.10\,(89.6\%)$ | $-0.37 \pm 0.16\,(82.9\%)$ | $-0.64 \pm 0.11\,(96.6\%)$ | $-0.36 \pm 0.11\,(81.0\%)$ |
| (15,10) | $-0.52 \pm 0.11\,(92.6\%)$ | $-0.39 \pm 0.08\,(91.5\%)$ | $-0.72 \pm 0.12\,(98.1\%)$ | $-0.51 \pm 0.16\,(90.6\%)$ |
| (15,15) | $-0.60 \pm 0.15\,(94.7\%)$ | $-0.44 \pm 0.11\,(91.6\%)$ | $-0.73 \pm 0.19\,(96.2\%)$ | $-0.65 \pm 0.13\,(95.9\%)$ |
| (25,7) | $-0.43 \pm 0.08\,(89.5\%)$ | $-0.34 \pm 0.13\,(83.0\%)$ | $-0.64 \pm 0.08\,(97.8\%)$ | $-0.26 \pm 0.11\,(72.6\%)$ |
| (25,10) | $-0.56 \pm 0.07\,(95.8\%)$ | $-0.44 \pm 0.09\,(94.2\%)$ | $-0.75 \pm 0.05\,(99.7\%)$ | $-0.44 \pm 0.13\,(87.5\%)$ |
| (25,15) | $-0.66 \pm 0.04\,(98.3\%)$ | $-0.49 \pm 0.08\,(94.9\%)$ | $-0.79 \pm 0.05\,(99.6\%)$ | $-0.64 \pm 0.13\,(96.1\%)$ |
| (50,7) | $-0.38 \pm 0.07\,(87.5\%)$ | $-0.46 \pm 0.07\,(91.2\%)$ | $-0.66 \pm 0.06\,(98.7\%)$ | $-0.22 \pm 0.08\,(68.4\%)$ |
| (50,10) | $-0.60 \pm 0.06\,(97.4\%)$ | $-0.53 \pm 0.04\,(97.3\%)$ | $-0.77 \pm 0.04\,(100\%)$ | $-0.35 \pm 0.09\,(81.2\%)$ |
| (50,15) | $-0.69 \pm 0.04\,(99.1\%)$ | $-0.55 \pm 0.05\,(97.5\%)$ | $-0.83 \pm 0.03\,(100\%)$ | $-0.64 \pm 0.08\,(97.9\%)$ |

*Table 9.* Average local Ricci evolution coefficients, computed using Forman Ricci curvature, on synthetic data. Values are means $\pm$ standard deviations over 50 independently trained networks per architecture; proportion of vertices with negative coefficients is reported in parentheses. Networks were randomly initialized.

| (Width,Depth) | Syn-I | Syn-II | Syn-III | Syn-IV |
|---|---|---|---|---|
| (15,7) | $-0.16 \pm 0.08\,(65.1\%)$ | $-0.23 \pm 0.11\,(70.9\%)$ | $-0.22 \pm 0.09\,(70.0\%)$ | $-0.09 \pm 0.08\,(58.0\%)$ |
| (15,10) | $-0.15 \pm 0.07\,(62.2\%)$ | $-0.19 \pm 0.08\,(71.9\%)$ | $-0.29 \pm 0.07\,(74.7\%)$ | $-0.17 \pm 0.10\,(65.3\%)$ |
| (15,15) | $-0.16 \pm 0.07\,(62.5\%)$ | $-0.16 \pm 0.10\,(66.2\%)$ | $-0.28 \pm 0.08\,(72.9\%)$ | $-0.22 \pm 0.17\,(67.7\%)$ |
| (25,7) | $-0.17 \pm 0.07\,(65.6\%)$ | $-0.26 \pm 0.08\,(74.0\%)$ | $-0.22 \pm 0.07\,(70.2\%)$ | $-0.02 \pm 0.07\,(51.6\%)$ |
| (25,10) | $-0.16 \pm 0.06\,(63.9\%)$ | $-0.26 \pm 0.06\,(75.2\%)$ | $-0.25 \pm 0.08\,(72.6\%)$ | $-0.11 \pm 0.09\,(60.3\%)$ |
| (25,15) | $-0.17 \pm 0.06\,(62.1\%)$ | $-0.22 \pm 0.06\,(71.5\%)$ | $-0.28 \pm 0.07\,(73.4\%)$ | $-0.26 \pm 0.06\,(71.7\%)$ |
| (50,7) | $-0.20 \pm 0.04\,(67.4\%)$ | $-0.32 \pm 0.07\,(78.1\%)$ | $-0.21 \pm 0.07\,(71.0\%)$ | $+0.02 \pm 0.05\,(46.5\%)$ |
| (50,10) | $-0.19 \pm 0.07\,(64.6\%)$ | $-0.35 \pm 0.05\,(79.2\%)$ | $-0.27 \pm 0.05\,(74.7\%)$ | $-0.04 \pm 0.05\,(53.6\%)$ |
| (50,15) | $-0.16 \pm 0.07\,(60.7\%)$ | $-0.30 \pm 0.05\,(75.2\%)$ | $-0.28 \pm 0.06\,(73.2\%)$ | $-0.24 \pm 0.06\,(71.1\%)$ |

### A.4.5. DOUBLE DESCENT PHENOMENON

In modern machine learning, it is common to train extremely large and heavily overparameterized models that achieve zero training error while still exhibiting strong generalization performance. This surprising behavior is captured by the *double descent* phenomenon, introduced by Belkin et al. (2019), which refines the classical view of the bias–variance trade-off. Whereas the traditional theory predicts a U-shaped generalization curve as model capacity increases, double descent reveals an additional regime: once the interpolation threshold is crossed, generalization error can decrease again with increasing capacity. Recent work has shown that this phenomenon is a fundamental property of overparameterized models, appearing across a wide range of settings including neural networks, ensemble methods, decision trees, and classical linear regression (Belkin et al., 2019; Ba et al., 2020; Deng et al., 2022).

Several explanations have been proposed for this behavior. One line of reasoning suggests that enlarging the function class increases the number of interpolating solutions, thereby making it more likely to find functions that not only fit the data but also exhibit higher smoothness and regularity. Such simpler solutions are favored by an implicit form of Occam's razor, indicating that overparameterization can promote generalization by biasing learning toward these low-complexity explanations (Belkin et al., 2019).

A promising direction for future work is to investigate the double descent phenomenon through the lens of local Ricci evolution coefficients. In the overparameterized regime, double descent suggests that further enlarging the network should lead to improved generalization. Our experiments show that increasing network size—either by adding depth at fixed width or by expanding width at fixed depth—systematically increases the proportion of vertices with negative Ricci coefficients. Figure 12 illustrates these findings on real-world datasets using neural networks with a fixed width of 50 neurons per layer while varying the depth. This observation indicates that larger models exhibit curvature-driven dynamics on a more global scale, potentially enabling them to capture the underlying geometry of the problem more effectively. Since models that better align with data geometry are expected to generalize better, this perspective highlights a potential interplay between capacity growth and geometric representation, offering a novel geometric perspective on the double descent phenomenon.

*Table 10.* Average local Ricci evolution coefficients on real-world data computed using the Spearman correlation. Values are means ± standard deviations over 50 independently trained networks per architecture; proportion of vertices with negative coefficients is reported in parentheses. Networks were randomly initialized.

| (Width,Depth) | MNIST-1v7 | MNIST-6v9 | FMNIST-SvS | FMNIST-SvD | CIFAR |
|---|---|---|---|---|---|
| $(15, 7)$ | $-0.51 \pm 0.08\,(89.8\%)$ | $-0.44 \pm 0.12\,(84.5\%)$ | $-0.39 \pm 0.06\,(82.1\%)$ | $-0.27 \pm 0.06\,(74.0\%)$ | $-0.40 \pm 0.11\,(85.4\%)$ |
| $(15, 10)$ | $-0.52 \pm 0.06\,(90.8\%)$ | $-0.51 \pm 0.07\,(90.2\%)$ | $-0.36 \pm 0.08\,(82.8\%)$ | $-0.31 \pm 0.09\,(80.1\%)$ | $-0.39 \pm 0.16\,(85.8\%)$ |
| $(15, 15)$ | $-0.36 \pm 0.08\,(88.6\%)$ | $-0.48 \pm 0.05\,(87.7\%)$ | $-0.51 \pm 0.10\,(92.4\%)$ | $-0.42 \pm 0.10\,(89.0\%)$ | $-0.55 \pm 0.19\,(92.9\%)$ |
| $(25, 7)$ | $-0.50 \pm 0.09\,(88.6\%)$ | $-0.45 \pm 0.11\,(85.1\%)$ | $-0.36 \pm 0.04\,(79.0\%)$ | $-0.27 \pm 0.06\,(74.2\%)$ | $-0.45 \pm 0.12\,(89.0\%)$ |
| $(25, 10)$ | $-0.53 \pm 0.07\,(91.1\%)$ | $-0.54 \pm 0.08\,(90.8\%)$ | $-0.36 \pm 0.08\,(82.5\%)$ | $-0.31 \pm 0.08\,(80.8\%)$ | $-0.55 \pm 0.14\,(94.1\%)$ |
| $(25, 15)$ | $-0.49 \pm 0.04\,(88.5\%)$ | $-0.48 \pm 0.06\,(87.9\%)$ | $-0.49 \pm 0.09\,(92.6\%)$ | $-0.47 \pm 0.10\,(91.2\%)$ | $-0.68 \pm 0.05\,(96.9\%)$ |
| $(50, 7)$ | $-0.60 \pm 0.07\,(93.1\%)$ | $-0.45 \pm 0.11\,(84.1\%)$ | $-0.37 \pm 0.05\,(79.9\%)$ | $-0.33 \pm 0.08\,(80.1\%)$ | $-0.52 \pm 0.12\,(93.1\%)$ |
| $(50, 10)$ | $-0.58 \pm 0.07\,(93.1\%)$ | $-0.59 \pm 0.09\,(92.9\%)$ | $-0.39 \pm 0.10\,(84.2\%)$ | $-0.43 \pm 0.10\,(88.4\%)$ | $-0.70 \pm 0.04\,(97.9\%)$ |
| $(50, 15)$ | $-0.53 \pm 0.03\,(89.9\%)$ | $-0.55 \pm 0.05\,(90.6\%)$ | $-0.56 \pm 0.06\,(94.6\%)$ | $-0.55 \pm 0.07\,(92.7\%)$ | $-0.71 \pm 0.03\,(97.1\%)$ |

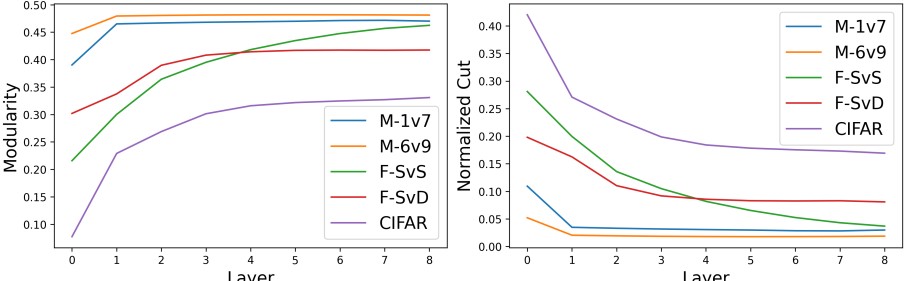

*Figure 10.* Modularity and normalized cut across network layers on real-world datasets. Reported values are averaged over 50 independently trained networks with random initialization.

## A.5. Details on experimental setup

All experiments were implemented in Python. Neural networks were built using PyTorch (v2.7.1). Default initialization schemes were used for the initial network weights. Networks were trained with binary cross-entropy loss and optimized using the standard Adam optimizer (Kinga et al., 2015) with a learning rate of 0.001. To solve the optimal transport problems required for computing Ollivier–Ricci curvature, we relied on the POT Python Optimal Transport library (v0.9.5). For constructing $k$-nearest neighbor graphs we used scikit-learn (v1.7.1), and for computing classical community strength measures such as modularity we employed NetworkX (v3.5). All figures in the main text were generated using Matplotlib (v3.10.5).

Our experiments were conducted on a local server with the specifications presented in Table 11.

*Table 11.* Hardware specifications.

| Components | Specifications |
|---|---|
| ARCHITECTURE | X86_64 |
| OS | Rocky Linux 8.10 (Green Obsidian) |
| CPU | Intel Xeon Platinum 8480CL 56-Core (×2) |
| GPU | NVIDIA H200 Tensor Core |
| RAM | 40GB |

We evaluate our approach on both synthetic and real-world datasets. The synthetic datasets, presented in Figure 13, are designed to exhibit different degrees of geometric and topological complexity, providing controlled settings to study curvature dynamics. For real-world data, we consider three benchmarks. MNIST (LeCun, 1998) consists of $28 \times 28$ grayscale images of handwritten digits (0–9). We focus on visually similar digit pairs, i.e., 1 vs. 7 (MNIST-1v7) and 6 vs. 9 (MNIST-6v9), to test the sensitivity of our approach to subtle shape differences. On Fashion-MNIST (Xiao et al., 2017), which contains grayscale images of clothing items, we consider sneakers vs. sandals (FMNIST-SvS) and shirts vs. dresses (FMNIST-SvD) as representative examples of fine-grained visual distinctions. Finally, on CIFAR-10 (Krizhevsky, 2009), a dataset of color natural images across ten object categories, we study cars vs. planes (CIFAR) as an example of two closely

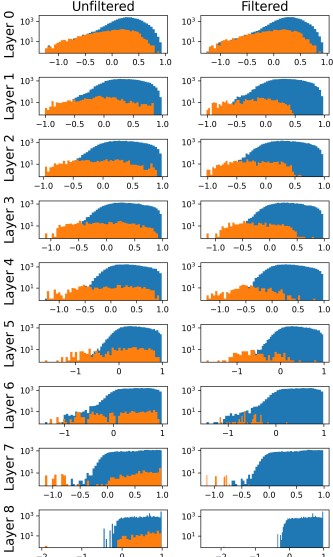

*Figure 11.* Curvature distributions for inter-community edges (orange) and intra-community edges (blue) on MNIST 1-vs-7 before (left column) and after (right column) removing misclassified samples.

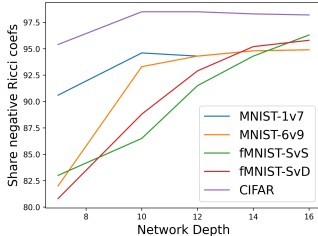

*Figure 12.* Proportion of vertices with negative local Ricci evolution coefficient for networks of varying depth.

related classes. Figure 14 illustrates representative samples from the real-world datasets.

Across all our experiments, we train the networks to achieve training accuracy above 99%, ensuring that our experiments evaluate meaningful learned feature representations. To verify that this reflects genuine generalization rather than overfitting, we evaluated all models on held-out test sets of 1,000 samples. The models achieve strong test accuracies between 95% and 99% across the synthetic and real datasets, with the exception of CIFAR, where test accuracy is slightly lower.

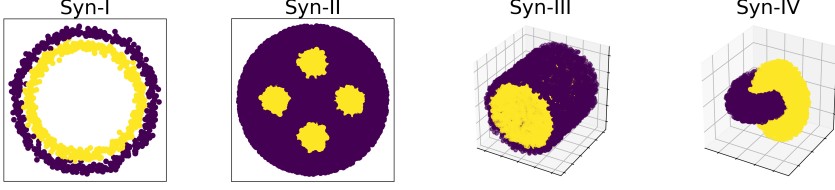

*Figure 13.* The synthetic datasets.

### A.5.1. HYPERPARAMETERS

The computation of local Ricci evolution coefficients requires constructing $k$-nearest neighbor graphs to approximate the geometry of the underlying manifold. The parameter $k$, which determines the number of neighbors each point connects to and thus controls the local scale of connectivity, plays a central role. Small values of $k$ capture fine-grained geometric structure but increase sensitivity to noise and may disconnect the graph. Larger values emphasize more global structure, at the cost of oversmoothing important local variations and raising the computational cost of Ollivier–Ricci curvature, which

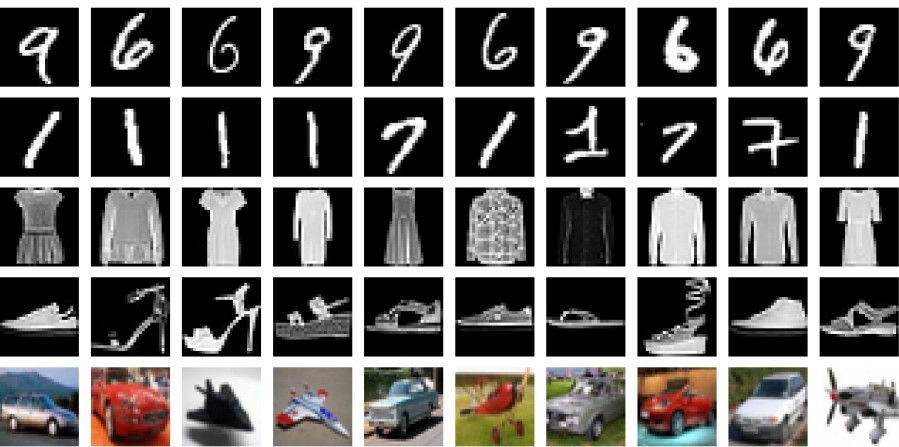

*Figure 14.* The real-world datasets.

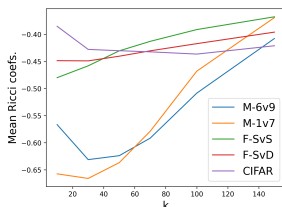

*Figure 15.* Mean local Ricci evolution coefficients for different neighborhood sizes $k$ on real datasets. Reported values are averaged over 50 independently trained networks.

scales cubically with the vertex degree. It is therefore not a priori clear how to choose $k$, as it mediates a fundamental trade-off between locality, robustness, and efficiency.

To investigate this trade-off, we conduct experiments across a range of neighborhood sizes. Specifically, we vary $k$ from 1% to 15% of the total size of the point cloud $X$, and compute the local Ricci evolution coefficients for each value of $k$. The resulting average coefficients on the real-world data are shown in Figure 15, where all reported values are averaged over 50 independently trained networks, each with width fixed to 50 and depth fixed to 10. We find that for small neighborhood sizes ($k$ between 1% and 5%), the local Ricci evolution coefficients remain relatively stable or even decrease. As $k$ increases further, the coefficients tend to rise, reflecting a weaker correlation between local Ricci curvature and the expansion or contraction of this region. This behavior is consistent across all datasets and across all network widths and depths examined in our experiments. Table 12 reports the local Ricci evolution coefficients for different neighborhood sizes and for different widths and depths on the MNIST-1v7 dataset. Across all architectures, we observe the same qualitative pattern as in Figure 15.

*Table 12.* Average local Ricci evolution coefficients on the MNIST-1v7 dataset across different neighborhood sizes $k$. Values are means $\pm$ standard deviations over 50 independently trained networks per architecture; proportion of vertices with negative coefficients is reported in parentheses. Networks were randomly initialized.

| (Width,Depth) | $k = 10$ | $k = 30$ | $k = 50$ | $k = 70$ | $k = 100$ |
|---|---|---|---|---|---|
| $(15, 7)$ | $-0.61 \pm 0.07\,(93.4\%)$ | $-0.61 \pm 0.07\,(91.7\%)$ | $-0.58 \pm 0.08\,(88.7\%)$ | $-0.53 \pm 0.07\,(83.9\%)$ | $-0.43 \pm 0.06\,(76.6\%)$ |
| $(15, 10)$ | $-0.63 \pm 0.05\,(96.6\%)$ | $-0.65 \pm 0.07\,(95.4\%)$ | $-0.60 \pm 0.06\,(91.8\%)$ | $-0.56 \pm 0.05\,(88.0\%)$ | $-0.45 \pm 0.05\,(79.0\%)$ |
| $(15, 15)$ | $-0.66 \pm 0.06\,(98.1\%)$ | $-0.62 \pm 0.17\,(95.1\%)$ | $-0.61 \pm 0.07\,(93.3\%)$ | $-0.55 \pm 0.08\,(88.1.0\%)$ | $-0.41 \pm 0.09\,(78.6\%)$ |
| $(25, 7)$ | $-0.60 \pm 0.06\,(93.3\%)$ | $-0.59 \pm 0.07\,(91.0\%)$ | $-0.58 \pm 0.05\,(89.2\%)$ | $-0.52 \pm 0.05\,(84.4\%)$ | $-0.44 \pm 0.05\,(77.2\%)$ |
| $(25, 10)$ | $-0.64 \pm 0.04\,(96.8\%)$ | $-0.64 \pm 0.05\,(95.5\%)$ | $-0.62 \pm 0.05\,(92.8\%)$ | $-0.57 \pm 0.05\,(88.4\%)$ | $-0.45 \pm 0.03\,(79.2\%)$ |
| $(25, 15)$ | $-0.65 \pm 0.04\,(98.2\%)$ | $-0.65 \pm 0.06\,(97.0\%)$ | $-0.60 \pm 0.06\,(94.2\%)$ | $-0.55 \pm 0.05\,(89.1\%)$ | $-0.43 \pm 0.04\,(79.6\%)$ |
| $(50, 7)$ | $-0.58 \pm 0.07\,(92.3\%)$ | $-0.62 \pm 0.05\,(93.3\%)$ | $-0.59 \pm 0.05\,(90.5\%)$ | $-0.55 \pm 0.05\,(86.1\%)$ | $-0.46 \pm 0.05\,(78.3\%)$ |
| $(50, 10)$ | $-0.65 \pm 0.05\,(97.6\%)$ | $-0.67 \pm 0.05\,(96.9\%)$ | $-0.65 \pm 0.04\,(94.6\%)$ | $-0.58 \pm 0.05\,(89.7\%)$ | $-0.47 \pm 0.04\,(80.4\%)$ |
| $(50, 15)$ | $-0.66 \pm 0.05\,(98.3\%)$ | $-0.64 \pm 0.06\,(97.3\%)$ | $-0.63 \pm 0.06\,(95.2\%)$ | $-0.55 \pm 0.05\,(90.4\%)$ | $-0.43 \pm 0.04\,(80.2\%)$ |

This behavior is expected, since we are approximating local geometric properties of the manifold using $k$-nearest neighbor graphs. When the neighborhood scale becomes too large, the one-hop neighborhoods of these graphs no longer correspond

to genuinely local regions of the manifold. Consequently, we expect a weaker correlation between the two quantities, as they cease to reflect the local nature of the Ricci flow.

To balance these effects, we fix $k = 5\%$ of the total size of the point cloud in the experiments reported in the main text. We additionally repeated the same experiments with $k = 3\%$ and $k = 7\%$, and observed quantitatively similar outcomes, showing that our findings are robust with respect to the precise choice of neighborhood size.

### A.5.2. LICENSES

We summarize the licenses of all code and datasets used in our experiments in Table 13.

*Table 13.* Licenses of code and datasets.

| Model/Dataset | License |
|---|---|
| MNIST (LeCun, 1998) | CC BY-SA 3.0 |
| Fashion-MNIST (Xiao et al., 2017) | MIT |
| CIFAR-10 (Krizhevsky, 2009) | MIT |
| PyTorch (Paszke et al., 2019) | 3-clause BSD |
| Scikit-learn (Pedregosa et al., 2011) | 3-clause BSD |
| POT (Python Optimal Transport) (Flamary et al., 2021) | 3-clause BSD |
| NetworkX (Hagberg et al., 2008) | 3-clause BSD |
| SciPy (Virtanen et al., 2020) | 3-clause BSD |

### A.6. LLM usage disclosure

We used an LLM during paper writing to improve grammar and wording.

