# OpenReview forum: "Neural Feature Geometry Evolves as Discrete Ricci Flow"
_ICML.cc/2026/Conference — ICML 2026 spotlight_

### Official Review · Reviewer_Aw7h · 2026-02-26

**Soundness:** 3
**Presentation:** 4
**Significance:** 3
**Originality:** 3
**Overall Recommendation:** 5
**Confidence:** 3

**Summary:**

This paper presents a framework to describe the evolution of feature geometry in neural networks via Ricci Flow.
The discrete Ricci curvature is computed on the k-nearest neighbor graph (or the r-distance neighboorhood graph) of the dataset or estimated using any method of choice.
Then the evolution of the Ricci curvature throughout the network is captured by doing the same for the features at each intermediate layer of the neural network.
The authors first show that wide enough linear layers, just after random initialization or trained for a fixed number of steps, do not change the graph structure (and thus the curvature) with high probability (Theorem 3.1 \& 3.3), but non-linearities do (Theorem 3.4).
Next the paper introduces (layerwise and network-wide) local Ricci evolution coefficients, which describe how feature distance evolution is correlated with discrete Ricci curvature.
Specifically, in Ricci flow, positive curvature results in a contraction in distances and vice-versa, thus a negative local Ricci evolution coefficient suggests feature evolution analogous to Ricci flow.
In extensive experiments the authors show that feed-forward classifiers on various datasets consistently show negative local Ricci evolution coefficients, suggesting an analogy between the evolution of their features and Ricci flow. They show that measures of community structure (modularity, normalized cut, curvature gap) are pronounced throughout the feature evolution.
Finally, they show how local Ricci evolution coefficients evolve during training, and with increasing network depth. In both cases a point of saturation is achieved, after which the coefficients increase, corresponding to the optimal number of layers in the latter case.

**Compliance With Llm Reviewing Policy:**

Affirmed.

**Final Justification:**

The paper presents a technically sound, well motivated analysis of neural feature geometry. I especially find the experimental design very logical and thorough, with the inclusion of the additional results that negative local Ricci evolution coefficients do not appear in meaningless neural features. Therefore I choose to keep my original recommendation to accept.

**Key Questions For Authors:**

1. You state that all of your models achieve training accuracy above 99\%, do they also achieve good accuracies on test data? Did you check for overfitting? What is the size of the test data set to compute the local Ricci evolution coefficients on?

2. How do local Ricci evolution coefficients look when you compute them for overfitted models and for a synthetic dataset where the classes cannot be separated?

3. How does the efficiency of computing discrete Ricci curvature scale to higher-dimensional data? Specifically, do you expect the number of datapoints needed for a sufficiently representative k-NN graph to scale exponentially with ambient or intrinsic dimension (or even lower)?

**Limitations:**

yes

**Strengths And Weaknesses:**

## Soundness:

The notion of analyzing the evolution of neural feature geometry through the lens of Ricci flow is well motivated by the authors.
While the settings are heavily simplified, Theorems 3.1 - 3.4 indicate the ability to change the neighborhood graphs may be connected to the performance of neural network architectures, though they do not show any causal relationship between the two. This is also confirmed experimentally.
The practical estimation of discrete Ricci curvature presents a trade-off between efficiency and accuracy, which the authors describe and explore very well in the paper.
The experiments are designed in a meaningful way:
The claim that neural feature geometry evolves similar to Ricci flow is supported by Table 1, the relevance of neighborhood graphs and community structures for neural network features is supported by figures 2 and 3, and practical implications  of the observed effects for neural network design are explored in figures 4 and 5.
With decreasing connections between the communities of both classes, outlier edges (e.g. from misclassified samples) can dominate community measures like curvature gap. Local Ricci evolution coefficients however seem robust across multiple different settings and hyperparameter choices in building neighborhood graphs (table 1, figure 14).
This paper show that the evolution of "good" neural feature geometry correlates with Ricci-Flow, suggesting that it may be a source or explanation for the performance of neural networks.
One experiment that I would like to see for completeness is testing whether "bad" neural feature geometry (overfitted models, data with no clear class separation) does *not* correlate with Ricci-Flow. Table 3 already gives a good impression for uninitialized models, including the other mentioned scenarios would provide a more complete picture that negative local Ricci evolution coefficients do not occur naturally.

## Presentation:

The structure and writing of this paper seem very polished. The presentation of the background is complete and logical, theorems are simplified to the necessary degree in the main text and set into proper context. The results are presented in a understandable way, though I feel like table 1 could be summarized more nicely by a fitting graphic. The appendix is complete and gives detailed follow-up on experiments, though I would like some details about the datasets and the performance of the classifiers on them (especially how high the test accuracy of the models is, and how big the test set is with which to compute the local Ricci evolution coefficients).

## Significance:

This paper helps to further the understanding of neural feature geometry and shows implications on performance and model design on real-world data. The exploration of this effect in other architectures (convolutional nets, transformers), other training paradigms (generative models, self-supervised representation learning) and differently structured data is left for future work. This work provides a solid foundation for such future work, with logical design choices.

## Originality:

The fundamental idea of this paper is very similar to the mentioned Baptista et al. (2024), meaning it is not entirely unique. That said, the authors do an excellent job in appendix A.1.1 elaborating on the exact differences and advantages of their formulation.

---

> ### Author Rebuttal · Authors · 2026-03-29
>
> We thank the reviewer for the encouraging and detailed feedback.
>
> Regarding Question 1: Thank you for raising this point, we will add these details to the revised paper in the experimental setup section. The test sets consist of 1.000 samples. We did check for overfitting, and the models achieved strong test accuracies between 95-99% across synthetic and real datasets, with the only exception being CIFAR where we obtained slightly lower test accuracies.
>
> Regarding Question 2: We ran additional experiments on synthetic data without underlying geometric structure, where the classes cannot be separated. As an example, we sampled points uniformly from the unit disc and assigned random labels, ensuring no meaningful class boundary exists. We trained neural networks on this task and computed the local Ricci evolution coefficients. As expected, the coefficients remain close to zero, indicating that the network does not learn geometric transformations when no separable structure is present. We believe that this validates our framework: the negative Ricci evolution coefficients observed in our main experiments do not occur naturally but genuinely reflect the geometric restructuring that arises when the network learns geometric structure.
>
> Regarding Overfitted models: We have analyzed the evolution of Ricci evolution coefficients in overfitted models throughout training and observe a consistent pattern across all experiments. During the initial phase of training, while test accuracy improves, Layer-Ricci coefficients decrease sharply, a behavior we attribute to the network extracting meaningful geometric structure from the data. Subsequently, as test accuracy stagnates or degrades, the Layer-Ricci coefficients begin to increase again. We refer the reviewer to Section 4.3 and to the response to Reviewer qHqB for more details.
>
> Regarding Question 3: The computational cost of discrete Ollivier-Ricci curvature depends on the $k$-NN graph structure, not the ambient dimension. But the reviewer raises an important related point, regarding the construction of a representative $k$-NN graph, which requires sufficient sampling density relative to the intrinsic dimension. The key theoretical requirement of our framework is consistency between continuous and discrete Ricci curvature, for which Trillos & Weber (2023) provide non-asymptotic guarantees whose sample complexity scales with intrinsic rather than ambient dimension. Since the intrinsic dimensionality of intermediate feature representations is typically much lower than the ambient dimension, and has been observed to decrease in deeper layers (Ansuini et al., 2019), we expect these scaling concerns to be mitigated in practice.

---

> > ### Author Rebuttal · Reviewer_Aw7h · 2026-04-01
> >
> > I thank the authors for answering my questions. The answers confirm my positive impression of the paper and I will keep my recommendation to accept.

---

### Official Review · Reviewer_9EAc · 2026-03-09

**Soundness:** 3
**Presentation:** 4
**Significance:** 2
**Originality:** 3
**Overall Recommendation:** 5
**Confidence:** 4

**Summary:**

The work investigates the evolution of the data manifold (discretized as a graph) through the layers of a deep network, both theoretically and empirically, and relates it to the (discrete) Ricci flow.

**Compliance With Llm Reviewing Policy:**

Affirmed.

**Final Justification:**

I find the paper original and sound, in both its theoretical and empirical aspects. It is also well-presented. I raised some concerns in the review, that were addressed during the rebuttal. This led me to increase my score. Thus, I believe that the paper deserves acceptance.

**Key Questions For Authors:**

Could you please elaborate on the assumption around $\epsilon$ in Theorem 3.1? What does it mean intuitively, in terms of the geometry of the point-cloud $X$? How “likely” is such an $\epsilon$ to exist?

**Limitations:**

Yes

**Strengths And Weaknesses:**

## Strengths

- The Ricci flow perspective on neural representations is mathematically original  and interesting. Even though, as mentioned in the paper, such a perspective has been previously proposed by Baptista et al. (2024), this work develops it significantly in several aspects.

- The paper provides an extensive experimentation, based on novel curvature-based metrics.

- The paper is remarkably well written. The exposition is clean and clear. The mathematical formalism is concise and precise, with coherent notation.

## Weaknesses

As a core weakness, I find the main theoretical results not particularly deep. The statements of the theorems are rather expected, and the proofs consist of a straightforward derivation from previous well-known results. Specifically:

- **Theorem 3.1**: As mentioned in the paper, Theorem 3.1 follows almost immediately from the Johnson-Lindenstrauss Lemma, which concerns approximate preservation of distances under random matrices (from which it is natural to expect preservation of neighbor graphs). The proof in appendix A.3.1 (lines 1011-1042) is, indeed, a simple derivation. The catch is that the assumption of Theorem 3.1 (i.e., the existence of $\epsilon$) is quite technical, and not discussed in the paper (see question below around this).

- **Theorem 3.3**: Similarly, this result follows from the fact that the (first-layer) weight matrices of wide networks stay close to initialization during gradient descent (this can be deduced, as done in the paper, from Du et. al (2018), but is also well-known, e.g., in the NTK literature). From here, the preservation of the neighbor graph follows by straightforward derivations similar to the ones from Theorem 3.1 (lines 1210-1263).

Moreover, I think that there is a discrepancy between the overall message of the paper, and the theoretical results. As stated in the introduction, the core message of the paper is to relate the evolution of the data manifold/graph through the network’s layers with the Ricci flow (see Figure 1). Indeed, this is extensively explored in the experiments (Section 4). However, the theoretical results do not directly contribute to this message. As mentioned above, Theorem 3.1 and 3.3 are “negative results” in this sense: they show, essentially, that (wide) linear maps/networks preserve the graph structure, and thus can not implement any (discrete) Ricci flow. The only “positive” theoretical result is Theorem 3.4, showing that a shallow ReLU network is able to change the geometry of the k-NN graph. This is a high-level result, indicating benefits of nonlinearities in terms of graph evolution. It does not provide any insight into the actual graph/feature evolution occurring in neural networks, nor draws any concrete connection to the Ricci flow.

Overall, I find the paper interesting and original. Thus, I am leaning towards acceptance. Yet, due to the limited depth and significance of the theoretical results, I am not strongly convinced.

---

> ### Author Rebuttal · Authors · 2026-03-29
>
> We thank the reviewer for carefully reading our paper and the encouraging feedback.
>
> Regarding Question 1: The condition ensures that, for each point $x$, there is a strict gap between the distance to its $k$-th and $(k+1)$-th nearest neighbors, so that neighborhood membership is unambiguous. The ratio $(1-\varepsilon)/(1+\varepsilon)$ measures the size of this gap: larger $\varepsilon$ reflects a more robust separation. This condition is generically satisfied in practice: when points are drawn from a continuous distribution, distance ties occur with probability zero, so a valid $\varepsilon > 0$ always exists. We will include this clarification in the revision.
>
> Regarding Weaknesses: We appreciate this critique and agree that our theoretical results do not explain why neural networks exhibit Ricci flow-like dynamics; providing such an explanation is a very challenging and interesting open problem. However, we would like to clarify the intended role of Section 3.1: The primary purpose is not to prove that neural networks must behave this way, but rather to establish why this behavior is possible from a mathematical standpoint. We believe that these results provide the mathematical foundation for interpreting our empirical findings and place the experiments in proper context. The theoretical and empirical contributions are thus complementary rather than explanatory of each other. We will revise the text to make this distinction more explicit.

---

> > ### Author Rebuttal · Reviewer_9EAc · 2026-04-01
> >
> > I thank the authors for their reply, which resolved my concerns. I will raise my score.

---

### Official Review · Reviewer_pHqB · 2026-03-11

**Soundness:** 3
**Presentation:** 4
**Significance:** 3
**Originality:** 4
**Overall Recommendation:** 5
**Confidence:** 4

**Summary:**

This paper studies how feature geometry evolves with deep neural network layers. Firstly, the authors show that random linear transformation maintains nearest neighbor structures in the wide regime; moreover,  the change of feature geometry is attributed to nonlinear activations. Secondly, the authors propose Ricci evolution coefficients to quantify how feature geometry evolves similarly to a Ricci flow. Lastly, the authors implement experiments to show Ricci-like revolution of feature geometry. Further, they study (1) the emergence of community structures as layer goes deeper; (2) how Ricci coefficients correspond to training time; (3) how Ricci coefficients correspond to layer depth.

**Compliance With Llm Reviewing Policy:**

Affirmed.

**Final Justification:**

This is an interesting paper and I originally gave a positive score. The authors' rebuttal reinforced my recommendation for acceptance.

**Key Questions For Authors:**

1.	The result in Figure 5 looks promising as the lowest in the left subfigure coincide with the highest in the right subfigure. It is known that ResNet-like network can benefit from depth. Can you discuss the phenomenon when you replace the network with a simple ResNet? Will the Ricci coefficient keep decreasing for a large number of layers?

2.	[See W1] Can you give an example where test accuracy decreases while training accuracy keeps improving, which would better support the proposed early-stopping interpretation?

3.	[See W2] How reasonable is it to consider the scenario where the change of weight matrix is very tiny with epochs?

**Limitations:**

yes

**Strengths And Weaknesses:**

Strengths:

1.	This paper analyzes the contribution of linear transformation and nonlinear activation of neural networks. The theoretical parts are sound and natural: Section 3.1 first addresses “why” feature geometry would change. Section 3.2 then tries to address “how” feature geometry would change.

2.	This paper is well-written and self-contained. It is well-structured. “Summary of contributions” clearly link contributions to each subsection. Appendix A.2 clearly presents the necessary background in a way accessible to a wide audience.

3.	I believe this work could well influence further research. For instance, similar methods could be applied to neural networks specifically designed for geometric data (e.g., “rewiring” could be useful in graph neural networks). Also, it introduces geometry-based methods to generic neural networks, which should influence a wide audience.

4.	The idea of local Ricci evolution coefficients is very interesting and novel to my best knowledge. It gives insights into how neural networks interact with feature geometry and will be useful for understanding deep learning.

Weaknesses:

1.	Section 4.3 discusses overfitting and Ricci coefficients. However, I don’t observe “overfitting” in Figure 4. The test accuracy keeps improving, though not that significant. So I doubt whether the authors should conclude that the network “begins to overfit individual samples”. Therefore, I am concerned about whether the Ricci coefficients could be used as a criterion for early stopping.

2.	I also have some concerns about the theoretic results in Section 3.1. The approach to proving Theorem 3.3 involves Corollary A.13, which basically says that “the change of weight matrix is very tiny”, and that is the reason why trained networks still provide graph isomorphisms. In practice, this would ask the neural network to stay close to being random and does not seem very reasonable.

3.	(Minor presentation issue) It would be useful to write out explicitly the “discrete Ricci flow” equation. Line 897 is a discretization in time. Is there something concrete for (manifold -> graph) discretization? Or one should only understand it as implicitly existing so that the coefficients are the best subject to study?

4.	(Minor presentation issue) Theorem 3.1: $Y \subset X$ should be $Y \subset X - \\\{x\\\}$.

---

> ### Author Rebuttal · Authors · 2026-03-29
>
> We thank the reviewer for the encouraging and detailed review.
>
> Regarding Question 1: We find this idea very interesting and have run additional experiments on ResNets using the same setup as in the paper (same datasets, random initialization, averaged over 50 runs). First, we note that we observe the same behavior for ResNets:  Layer-Ricci coefficients are negative and follow the same trend across network depths. Furthermore, the results confirm the reviewer's conjecture: Layer-Ricci coefficients in ResNets decrease for a larger number of layers. A comparison of Layer-Ricci evolution coefficients for ResNets and standard DNNs on the MNIST 1-vs-7 dataset is provided at the following link:
>
> https://anonymous.4open.science/r/rebuttal_plots-DD66/Layer_Ricci_Coefficients_resnet.png
>
> We find these results compelling and will incorporate them into the revised paper. Thank you for bringing up this idea.
>
> Regarding Question 2: We thank the reviewer for this fair and constructive observation. To address this, we conducted additional experiments on the synthetic dataset Syn-I in which we flipped 25% of the training labels, creating a setting where genuine overfitting occurs. We observe behavior consistent with our interpretation: during the initial phase, as the model learns the true decision boundary, the local Ricci evolution coefficients exhibit a sharp decline, coinciding with test accuracy rising to approximately 95%. After this point, training accuracy continues to improve while test accuracy drops by 13.35%, a clear sign of overfitting to the corrupted labels. During this second phase, the Ricci evolution coefficients increase again, indicating that the network's geometric transformations are no longer as strongly aligned with curvature-driven dynamics. We will include these improved experiments in the revised manuscript. A plot of the test accuracies and the corresponding Ricci evolution coefficients can be found under the following link:
>
> https://anonymous.4open.science/r/rebuttal_plots-DD66/rebuttal_training_dynamics.png
>
> Regarding Question 3: The theoretical analysis in Theorem 3.3 is conducted within the lazy training regime, in which weights remain close to their initialization throughout training, a setting known to hold for sufficiently overparameterized models (Du et al., 2018). This is intentional: the result is designed to expose an inherent limitation of this regime, demonstrating that it is fundamentally constrained in its capacity for geometric transformation and that nonlinearity is essential for meaningful geometric change. These findings complement existing work on the limitations of the regime by providing a geometric characterization of what such networks cannot achieve. Theorem 3.4 then establishes that introducing a ReLU nonlinearity suffices to enable graph rewiring, a finding confirmed empirically in Section 4. We will make this framing explicit in the revised manuscript.
>
> Regarding Weakness 3: Thank you for this question. You are correct that the equation in line 897 is a temporal discretization and one of several proposed discrete Ricci flow formulations on graphs. As noted in lines 205–209 and 901–907, there is no canonical discrete Ricci flow: multiple formulations exist based on different curvature notions (Ollivier, Forman, Bakry-Émery) and update rules.
> Precisely because no single formulation is canonical, we do not commit to a specific flow equation and check whether networks follow it exactly. Instead, our local Ricci evolution coefficients (Section 3.2) are derived from the core mathematical intuition of continuous Ricci flow: positive curvature contracts, negative curvature expands, and measure whether this qualitative relationship holds between curvature and local distance changes across layers. The coefficients thus capture the signature of Ricci flow dynamics rather than tracking any particular discretization, making our framework agnostic to the choice of discrete formulation. We will clarify this explicitly in the revised manuscript.
>
> Regarding Weakness 4: Thank you, we will correct this in the revised paper.

---

> > ### Author Rebuttal · Reviewer_pHqB · 2026-04-02
> >
> > I thank the authors for the response. I keep my recommendation for acceptance.

---

### Official Review · Reviewer_tC3C · 2026-03-19

**Soundness:** 3
**Presentation:** 3
**Significance:** 3
**Originality:** 3
**Overall Recommendation:** 5
**Confidence:** 4

**Summary:**

The paper mainly provides an empirical exploration of the evolution of neural feature geometry from the point of view of discrete geometry. The (unknown) input data manifold (* see below for "manifold") is represented as a simple geometric graph, as are the points via the intermediate feature representations. This is done both for deep linear networks, and networks where you have pointwise non-linearities. A number of experimental tasks are chosen: 1 vs. 7, 6 vs. 9 on MNIST, sneakers vs. sandals on FMNIST, shirts vs. dresses on FMNIST,
and from CIFAR-10 (cars vs. planes). A total of 20,000 experiments are carried out (counting the fact that experiments are run multiple times). On these experiments, it is observed that the constructed geometric graphs evolve as a discrete Ricci flow (showing negative local Ricci evolution coefficient). Further, class separability is observed by the emergence of community structure in the graphs. Some basic theoretical results are also provided to support some of the claims (e.g. about the utility of non-linearities).

**Compliance With Llm Reviewing Policy:**

Affirmed.

**Key Questions For Authors:**

None

**Limitations:**

Yes

**Strengths And Weaknesses:**

Strengths: I think the paper's theoretical contribution is basic. However, the paper is essentially one in very few of its type (another is a cited paper by Natizat and co) that shows how the internal representations evolve. The perspective taken here is that of discrete geometry. The authors consider the work of Ollivier and from it consider simple measures that can help quantify the Ricci flow dynamics. The experiments show that many vague intuitions (and sometimes non-vague, see the work of Yarotsky on evolution of weights) that people have about NN layers' evolution during training can be tired to fairly specific and natural notions. The experiments are on basic tasks, but are numerous enough to be able to support the main claim sufficiently. They provide evidence that could use formalization and greater understanding in future work. I would like the paper to be published as a consequence.

Weaknesses: The theoretical results are kind of restatements of known results or are too basic. The difference is that some ideas from Ricci flow are connected to them. But in any case, they are necessary to place the experiments in proper context.

Not exactly a weakness, but one thing that unfortunately annoyed me about the paper is the loose usage of the word "manifold." Saying "data manifold" is a bit of a meme in ML and this paper seems to fully lean into the meme. I want to note that there is no way to know that the data lives on a manifold, or even an union of manifold. If you are using a NN which is not invertible, and you are using ReLU non-linearities, the intermediate spaces can never be manifolds (in fact not even sub-manifolds). It is much more defensible to say that natural data distributions live near a low-dimensional tame geometric object (which can sometimes be manfolds). Neural networks can represent functions whose images are also tame.

---

> ### Author Rebuttal · Authors · 2026-03-29
>
> We thank the reviewer for the encouraging review.
>
> Regarding the use of the term “manifold”: We acknowledge that the term manifold is used informally in the paper. We invoke the manifold assumption primarily as a heuristic motivation for applying discrete geometric tools like $k$-NN graphs and discrete Ollivier–Ricci curvature. However, our empirical results are properties of graphs constructed from finite point clouds and do not rely on the underlying data distribution being supported on a smooth manifold. We will revise the manuscript to adopt more precise terminology throughout and include an explicit clarification of this in the Discussion section.

---

> > ### Author Rebuttal · Reviewer_tC3C · 2026-04-04
> >
> > Thanks a lot for the detailed rebuttal (I read through the responses to all the reviewers). I will keep my (already positive) score.

---

### Decision · Program_Chairs · 2026-04-30

**Decision:**

Accept (spotlight)

**Comment:**

The authors study the evolution of neural feature representations under discrete geometry perspectives. The authors claim that feature geometry in deep neural networks evolves analogously to a discrete Ricci flow, which is a novel and compelling geometric perspective on representation learning. The authors provide extensive experiments across multiple datasets show consistent Ricci-flow-like behavior, connections to class separability via community structure, and correlations with training time and network depth.

A major strength of the submission is its original perspective, which links neural representation learning with discrete Ricci flow. However, the reviewers also raised concerns on the limited depth of the theoretical contributions, especially it lacks theoretical results to explain the main empirical finding, i.e., why neural networks exhibit Ricci-flow-like behavior, which leaves a gap between the main claim and its theoretical support.

Although there are limitations in its theoretical depth, we think that the submission offers a novel insight, potentially impactful contribution, particularly through its empirical findings and proposed geometric framework. Therefore, we recommend acceptance for the submission.